# Distinct amyloid fibril structures formed by ALS-causing SOD1 mutants G93A and D101N

Mu-Ya Zhang[1,10], Yeyang Ma[2,3,10], Li-Qiang Wang [1,10✉], Wencheng Xia[2,3], Xiang-Ning Li[4], Kun Zhao[2,3], Jie Chen[1], Dan Li[5,6], Liangyu Zou [7], Zhengzhi Wang[8], Cong Liu [2,9✉] & Yi Liang [1✉]

## Abstract

**Two hundred eight genetic mutations in SOD1 have been linked to amyotrophic lateral sclerosis (ALS). Of these, the G93A and D101N variants maintain much of their physiological function, closely resembling that of wild-type SOD1, and the SOD1-G93A transgenic mouse is the most extensively used mouse line in the study of ALS. In this study, we report two cryo-EM structures of amyloid fibrils formed by G93A and D101N mutants of SOD1 protein. These mutations give rise to amyloid fibrils with distinct structures compared to native SOD1 fibrils. The fibril core displays a serpentine configuration featuring four β-strands, held together by two hydrophobic cavities and a salt bridge between Arg143 and Asp96 in the G93A fibril, and by a hydrophobic cavity and a salt bridge between Arg143 and Asp132 in the D101N fibril, demonstrating unique structural features for each mutant. Moreover, our results show that G93A fibrils are significantly more toxic than those formed by D101N, which do not show a marked increase in toxicity compared to wild-type SOD1 fibrils. This study sheds light on the structural mechanisms through which SOD1 mutants aggregate and induce cytotoxicity in ALS.**

**Keywords** Amyotrophic Lateral Sclerosis (ALS); SOD1 Mutants; Amyloid Fibrils; Cryo-EM Structure; Toxicity
**Subject Categories** Molecular Biology of Disease; Neuroscience; Structural Biology

## Introduction

Amyotrophic lateral sclerosis (ALS), also called Lou Gehrig's disease, is a progressive, fatal neurodegenerative disease characterized by the selective death of motor neurons (Cleveland and Rothstein, 2001; Shaw and Valentine, 2007; Chattopadhyay and Valentine, 2009; Ayers and Borchelt, 2021; Miller et al, 2022; Mead et al, 2023). The mechanism by which motor neurons die in individuals with ALS remains a fundamental unanswered question. Approximately 90% of ALS cases are sporadic, while approximately 10% are familial and are typically inherited (Cleveland and Rothstein, 2001; Valentine et al, 2005; Shaw and Valentine, 2007; Chattopadhyay and Valentine, 2009; Polymenidou and Cleveland, 2011; Ajroud-Driss and Siddique, 2015; Ayers and Borchelt, 2021). Currently, more than 50 genes have been implicated in ALS pathogenesis (Ayers and Borchelt, 2021). Among these genes, the *sod1* gene, the first gene associated with a familial form of ALS (Rosen et al, 1993), is the second most common cause of the disease (Polymenidou and Cleveland, 2011; Ajroud-Driss and Siddique, 2015; Ayers and Borchelt, 2021). Approximately 2–6% of ALS cases are caused by mutations in the antioxidant enzyme Cu, Zn-superoxide dismutase (SOD1) (Rosen et al, 1993; Cleveland and Rothstein, 2001; Valentine et al, 2005; Shaw and Valentine, 2007; Polymenidou and Cleveland, 2011; Ajroud-Driss and Siddique, 2015; Ayers and Borchelt, 2021; Miller et al, 2022; Mead et al, 2023). Notably, two hundred eight genetic mutations in SOD1 have been identified in the familial form of ALS (Rosen et al, 1993; Ripps et al, 1995; Bruijn et al, 1997; Cudkowicz et al, 1997; Cleveland and Rothstein, 2001; Rodriguez et al, 2002; Arisato et al, 2003; Tiwari and Hayward, 2003; Tiwari et al, 2005; Valentine et al, 2005; Jonsson et al, 2006; Shaw and Valentine, 2007; Furukawa et al, 2008; Shaw et al, 2008; Wang et al, 2008; Chattopadhyay and Valentine, 2009; Prudencio et al, 2009; Furukawa et al, 2010; Münch et al, 2011; Chan et al, 2013; Ayers et al, 2014a; Nagano et al, 2015; Ayers et al, 2016; Sekhar et al, 2016; Sekhar and Kay, 2019; Ayers and Borchelt, 2021; Miller et al, 2022) (https://alsod.iop.kcl.ac.uk/). These mutations have remarkably diverse effects on the structure, activity, and stability of the native state of SOD1 (Rodriguez et al, 2002; Valentine et al, 2005; Shaw and Valentine, 2007; Furukawa et al, 2008; Wang et al, 2008; Chattopadhyay and Valentine, 2009; Prudencio et al, 2009; Nagano et al, 2015; Sekhar et al, 2016; Sekhar and Kay, 2019; Ayers and Borchelt, 2021). The cytoplasmic aggregation of these mutants and wild-type SOD1 in motor neurons is a pathological feature of ALS

[1]Hubei Key Laboratory of Cell Homeostasis, College of Life Sciences, TaiKang Center for Life and Medical Sciences, Wuhan University, 430072 Wuhan, China. [2]Interdisciplinary Research Center on Biology and Chemistry, State Key Laboratory of Chemical Biology, Shanghai Institute of Organic Chemistry, Chinese Academy of Sciences, 201210 Shanghai, China. [3]University of Chinese Academy of Sciences, 100049 Beijing, China. [4]The Cryo-EM Center, Core Facility of Wuhan University, Wuhan University, 430072 Wuhan, China. [5]Key Laboratory for the Genetics of Developmental and Neuropsychiatric Disorders, Ministry of Education, Bio-X Institutes, Shanghai Jiao Tong University, 200030 Shanghai, China. [6]Zhangjiang Institute for Advanced Study, Shanghai Jiao Tong University, 200240 Shanghai, China. [7]Department of Neurology, Shenzhen People's Hospital (the First Affiliated Hospital of Southern University of Science and Technology), the Second Clinical Medical College, Jinan University, 518020 Shenzhen, China. [8]School of Civil Engineering, Wuhan University, 430072 Wuhan, China. [9]Shanghai Academy of Natural Sciences (SANS), Fudan University, Shanghai, China. [10]These authors contributed equally: Mu-Ya Zhang, Yeyang Ma, Li-Qiang Wang. ✉E-mail: wangliqiang@whu.edu.cn; liulab@sioc.ac.cn; liangyi@whu.edu.cn

(Bruijn et al, 1997; Cleveland and Rothstein, 2001; Valentine et al, 2005; Jonsson et al, 2006; Shaw and Valentine, 2007; Furukawa et al, 2008; Shaw et al, 2008; Wang et al, 2008; Chattopadhyay and Valentine, 2009; Prudencio et al, 2009; Furukawa et al, 2010; Polymenidou and Cleveland, 2011; Chan et al, 2013; Ayers et al, 2014a; Ajroud-Driss and Siddique, 2015; Nagano et al, 2015; Ayers and Borchelt, 2021). It is, however, currently unknown whether different ALS-causing SOD1 mutations produce distinct SOD1 strains that influence the evolution of the disease (Bergh et al, 2015; Bidhendi et al, 2016; Ayers and Borchelt, 2021) and whether these mutations promote SOD1 aggregation through fundamentally distinct mechanisms (Bruijn et al, 1998; Cleveland and Rothstein, 2001; Rodriguez et al, 2002; Shaw and Valentine, 2007; Shaw et al, 2008; Wang et al, 2008; Chattopadhyay and Valentine, 2009; Prudencio et al, 2009; Furukawa et al, 2010; Münch et al, 2011; Chan et al, 2013; Ayers et al, 2014a; Bergh et al, 2015; Bidhendi et al, 2016; Ayers and Borchelt, 2021).

Since SOD1 was found to be associated with familial ALS in 1993 (Rosen et al, 1993), great efforts have been made to elucidate the atomic structure of SOD1 aggregates (Ivanova et al, 2014; Sangwan et al, 2017; Sangwan et al, 2018; Ayers and Borchelt, 2021; Iwakawa et al, 2021; Wang et al, 2022) and SOD1 strains (Bergh et al, 2015; Bidhendi et al, 2016; Ayers and Borchelt, 2021). Our recent cryo–electron microscopy (cryo-EM) structure of disulfide-reduced, apo, full-length wild-type human SOD1 amyloid fibrils revealed an in-register intramolecular β-strand architecture. This structure, stabilized by salt bridges, hydrophobic cavities, and hydrogen-bonding networks, elucidates the conversion of SOD1 from an immature form to an aggregated form during ALS pathogenesis (Wang et al, 2022). Additionally, cytoplasmic aggregation of transactive response DNA-binding protein-43 (TDP-43) in neurons constitutes another pathological hallmark of ALS (Arseni et al, 2022; Arseni et al, 2023; Sharma et al, 2024). Very recently, Sharma and co-workers reported cryo-EM structures of in vitro-generated amyloid fibrils from full-length TDP-43 (Sharma et al, 2024), which are different from those of TDP-43 amyloid fibrils purified from the brains of patients with ALS and frontotemporal lobar degeneration (Arseni et al, 2022; Arseni et al, 2023). However, despite three decades of investigation (Rosen et al, 1993; Ripps et al, 1995; Bruijn et al, 1997; Cudkowicz et al, 1997; Bruijn et al, 1998; Cleveland and Rothstein, 2001; Rodriguez et al, 2002; Arisato et al, 2003; Tiwari and Hayward, 2003; Tiwari et al, 2005; Valentine et al, 2005; Jonsson et al, 2006; Shaw and Valentine, 2007; Furukawa et al, 2008; Shaw et al, 2008; Wang et al, 2008; Chattopadhyay and Valentine, 2009; Prudencio et al, 2009; Furukawa et al, 2010; Münch et al, 2011; Polymenidou and Cleveland, 2011; Chan et al, 2013; Ayers et al, 2014a; Ivanova et al, 2014; Ajroud-Driss and Siddique, 2015; Bergh et al, 2015; Nagano et al, 2015; Ayers et al, 2016; Bidhendi et al, 2016; Sekhar et al, 2016; Sangwan et al, 2017; Sangwan et al, 2018; Sekhar and Kay, 2019; Ayers and Borchelt, 2021; Iwakawa et al, 2021; Arseni et al, 2022; Miller et al, 2022; Wang et al, 2022; Arseni et al, 2023; Sharma et al, 2024), atomic structural information on in vivo-derived SOD1 amyloid fibrils is not available and the molecular mechanisms by which mutations in SOD1 cause the familial form of ALS remain a mystery.

There are two subsets of mutations in SOD1 linked to the familial form of ALS. Metal-binding region mutants, such as H46R, H46D, G85R, D125H, and S134N, have mutations localized in and

around the metal-binding sites in SOD1 that substantially alter its biophysical properties relative to those of the wild-type protein (Rosen et al, 1993; Ripps et al, 1995; Bruijn et al, 1997; Bruijn et al, 1998; Cleveland and Rothstein, 2001; Rodriguez et al, 2002; Arisato et al, 2003; Tiwari and Hayward, 2003; Tiwari et al, 2005; Valentine et al, 2005; Jonsson et al, 2006; Shaw and Valentine, 2007; Furukawa et al, 2008; Shaw et al, 2008; Wang et al, 2008; Chattopadhyay and Valentine, 2009; Prudencio et al, 2009; Furukawa et al, 2010; Münch et al, 2011; Bergh et al, 2015; Nagano et al, 2015; Ayers et al, 2016; Bidhendi et al, 2016; Sekhar et al, 2016; Sekhar and Kay, 2019; Ayers and Borchelt, 2021; Miller et al, 2022; Wang et al, 2022). In contrast, wild-type-like mutants, such as A4V, D90A, G93A, D101G, and D101N, retain most of their physiological activities similar to those of wild-type SOD1 (Rosen et al, 1993; Cudkowicz et al, 1997; Cleveland and Rothstein, 2001; Rodriguez et al, 2002; Tiwari and Hayward, 2003; Tiwari et al, 2005; Valentine et al, 2005; Jonsson et al, 2006; Shaw and Valentine, 2007; Furukawa et al, 2008; Shaw et al, 2008; Wang et al, 2008; Chattopadhyay and Valentine, 2009; Prudencio et al, 2009; Furukawa et al, 2010; Chan et al, 2013; Ayers et al, 2014a; Bergh et al, 2015; Bidhendi et al, 2016; Sekhar et al, 2016; Sekhar and Kay, 2019; Ayers and Borchelt, 2021; Miller et al, 2022; Wang et al, 2022). In this study, we focused specifically on two ALS-causing SOD1 mutant proteins, G93A and D101N, for the following reasons. First, G93A and D101N are wild-type-like SOD1 mutants (Rosen et al, 1993; Cleveland and Rothstein, 2001; Rodriguez et al, 2002; Tiwari and Hayward, 2003; Tiwari et al, 2005; Valentine et al, 2005; Jonsson et al, 2006; Shaw and Valentine, 2007; Shaw et al, 2008; Wang et al, 2008; Chattopadhyay and Valentine, 2009; Prudencio et al, 2009; Furukawa et al, 2010; Chan et al, 2013; Ayers et al, 2014a; Bergh et al, 2015; Sekhar et al, 2016; Sekhar and Kay, 2019; Ayers and Borchelt, 2021; Wang et al, 2022); the aggregates or inclusions formed by G93A and D101N exhibit prion-like properties (Ayers et al, 2014a; Ayers et al, 2021; Ayers et al, 2023). Second, the SOD1-G93A transgenic mice, the most extensively used mouse line in the study of ALS, rapidly develop an ALS-like phenotype comprising paralysis and muscle loss at about 3 months of age (Gurney et al, 1994; Cleveland and Rothstein, 2001; Olsen et al, 2001; Jonsson et al, 2006; Ayers et al, 2014b; Zhang et al, 2014; Bergh et al, 2015; Ayers et al, 2021; Liu et al, 2024); and the D101N mutation is associated with rapidly progressing motor neuron degeneration (Ayers et al, 2014a; Ayers et al, 2021). However, the mechanisms underlying these phenomena remain unclear. Third, accelerating ALS progression in newborn G85R SOD1 transgenic mice by injecting in vitro-generated G93A or D101N amyloid fibrils establishes the disease relevance of these in vitro preparations (Ayers et al, 2016; Ayers et al, 2021). Fourth, although the wild-type SOD1 fibril structure reveals that familial mutations including D101N can disrupt key salt bridges in the cytotoxic form (Wang et al, 2022), whether G93A and D101N mutants form structurally distinct fibrils is still unknown.

Here, we report the cryo-EM–determined atomic structures of homogeneous amyloid fibrils assembled in vitro from apo, reduced forms of the ALS-causing SOD1 mutants G93A and D101N. We established that each mutant forms a unique fibril structure and that G93A fibrils exhibit markedly enhanced toxicity compared with wild-type SOD1 fibrils. This work provides mechanistic and structural insights into how SOD1 mutations drive cytotoxicity in familial ALS.

# Results

## The cryo-EM structures of the G93A fibril and the D101N fibril are compared with each other

To mimic physiological reducing conditions (Wang et al, 2022), apo forms of G93A and D101 SOD1 were first treated with 5 mM tris (2-carboxyethyl) phosphine (TCEP), a highly stable disulfide-reducing agent. Recombinant, full-length human apo SOD1 (residues 1−153) carrying either the G93A or D101N mutation, overexpressed in *Escherichia coli*, was then used for amyloid fibril formation. Purified apoproteins were incubated in 20 mM Tris-HCl buffer (pH 7.4) with 5 mM TCEP at 37 °C under agitation for 40–48 h (see "Methods"). Resulting amyloid fibrils were concentrated to ~30 μM using a centrifugal filter (Millipore) and then examined by electron microscopy.

Negative-staining transmission electron microscopy (TEM) revealed homogeneous and unbranched fibrils formed by apo-G93A and apo-D101N SOD1 under reducing conditions (Fig. EV1A,B). Atomic force microscopy (AFM) and cryo-EM were then used to compare the images (Fig. EV1C,D) and determine the atomic structures of these amyloid fibrils (Figs. 1–3; Table 1). The AFM images, cryo-EM micrographs, and two-dimensional (2D) class average images using RELION3.1 (Scheres, 2020) revealed that both G93A and D101N self-assembled amyloid fibrils made of a single protofilament with a left-handed helical twist (Figs. EV1C−F and EV2A,B). Both the G93A fibril and the D101N fibril were morphologically homogeneous but with different helical pitches (Fig. EV1C−F), and the G93A fibril exhibited a much longer helical pitch (243 ± 11 nm, Fig. EV1C) than the D101N fibril (64.3 ± 4.5 nm, Fig. EV1D).

Using helical reconstruction in RELION3.1 (Scheres, 2020), density maps of the ordered cores of the G93A fibril and the D101N fibril were determined to 3.09-Å and 2.92-Å resolution, respectively, featuring well-resolved side chain densities and clearly separated β strands along the fibril axis (Figs. 1A,B and EV3A,B). We then compared the three-dimensional (3D) maps of the here analyzed G93A and D101N SOD1 fibrils to previously published data on the wild-type SOD1 fibril (Wang et al, 2022). A cross-sectional view of the 3D map of the G93A fibril or the D101N fibril revealed a protofilament comprising a C-terminal segment (Fig. 1A,B). In sharp contrast, a cross-sectional view of the 3D map of the wild-type SOD1 fibril showed a protofilament containing both N- and C-terminal segments bridged by a flexible linker (Wang et al, 2022) (Fig. 1C). 3D maps of the G93A fibril and the D101N fibril revealed a single protofilament intertwined into a left-handed helix, with fibril core widths of ~8.1 nm and ~6.3 nm and helical pitches of 240.6 nm and 65.8 nm, respectively (Fig. 1D,E). The subunits within the G93A protofilament and the D101N protofilament stacked along the fibril axis with helical rises of 4.88 Å and 4.82 Å and twists of −0.73° and −2.63°, respectively (Fig. 1D,E). Collectively, these data demonstrated that under reducing conditions, bacterial-purified G93A and D101N formed different amyloid fibril structures.

## ALS-causing SOD1 mutant proteins G93A and D101N form distinct amyloid fibril structures

We unambiguously constructed a structural model of G93A fibrils comprising a C-terminal segment (residues 82–153) at 3.09 Å

(Fig. 2) and of D101N fibrils comprising a C-terminal segment (residues 95–153) at 2.92 Å (Fig. 3). The SDS–PAGE gels of SOD1 protein before (G93A monomer and D101N monomer) and after in vitro aggregation (G93A fibril and D101N fibril) have been added to show the intact of the protein (Appendix Fig. S1). The SDS–PAGE experiments showed that the protein was not degraded (Appendix Fig. S1) though no proteinase inhibitors were used during apo SOD1 protein purification, but still only C-terminal protein was incorporated in the ordered core of structures (Figs. 2 and 3). Side chain densities for most residues in the G93A fibril and many residues in the D101N fibril had high local resolution (3.0–3.2 Å) (Fig. EV3C,D). Using the approach by Amunts et al (2014), we compared the FSC curves between the final refined model and the map reconstructed from all fibrils, the FSC curves between a model refined against the first half of the two independent half maps used for gold-standard FSC *versus* the reconstruction from that same half, and the FSC curves between a model refined against the first half of the two independent half maps versus the second independent half map, and a good superimposition of the two independent halves was observed (Fig. EV4A,B). The vertical lines at 3.09 Å and 2.92 Å indicate the highest resolution used in model refinement of the G93A fibril and the D101N fibril, respectively (Fig. EV4A,B). These data convincingly demonstrate the absence of overfitting. For each structure of the G93A fibril and the D101N fibril, separate model refinements were performed against a single half-map, and the resulting model was compared with the other half-map to further confirm the absence of overfitting (Fig. EV4C–F). The comparison of the two optimized half-maps, half1 map and half2 map, from the G93A fibril (Fig. EV4C) and the D101N fibril (Fig. EV4D) 3D auto-refine process without mask demonstrates that the two half-maps match well. The side chains of the residues in the fibril cores of G93A and D101N can be well accommodated in the density maps (Figs. 2A and 3A). The exteriors of the fibril cores of G93A and D101N are mostly hydrophilic, carrying many negatively charged or positively charged residues, whereas the side chains of most hydrophobic residues are located mainly in the interiors of the G93A fibril fold and the D101N fibril fold except that the hydrophobic side chains of Ala95, Val97, Ile99, Ile104, and Leu106 are located in the exterior of D101N fibrils (Figs. 2B,F,G and 3B,F,G). Two hydrophobic cavities (Fig. 2B,G), a salt bridge (Fig. EV5A,B), two hydrogen bonds (Fig. EV5A,B), and a very compact fold (Fig. 2B,D) help stabilize the G93A fibril core, whereas a hydrophobic cavity (Fig. 3B,G), a salt bridge (Figs. 3F and EV6A,B), four hydrogen bonds (Fig. EV6A,C–E), and a less compact fold (Fig. 3B,D) help stabilize the D101N fibril core, as described in detail below.

The hydrophobic side chains of Ile104, Leu106, Ile113, Leu117, Val119, Ala123, Leu144, Ala145, Val148, Ile149, Ile151, and Ala152 are located in the interior of the G93A fibril (Fig. 2B) to form a stable hydrophobic core. The hydrophobic side chains of Leu84, Val87, Ala89, Ala93, Ala95, Val97, and Ile99 are located in the interior of the G93A fibril (Fig. 2B) to form the second hydrophobic cavity. In sharp contrast, the hydrophilic side chains of Asp96, Ser98, Glu100, Asn139, Arg143, and Gln153 are located in the interior of the G93A fibril and form a hydrophilic cavity (Fig. 2B). The hydrophobic side chains of Ile112, Leu117, Val119, Ala123, Leu127, Leu144, Val148, Ile149, Ile151, and Ala152 are located in the interior of the D101N fibril (Fig. 3B) to form a stable hydrophobic core. In sharp contrast, the hydrophilic side chains of

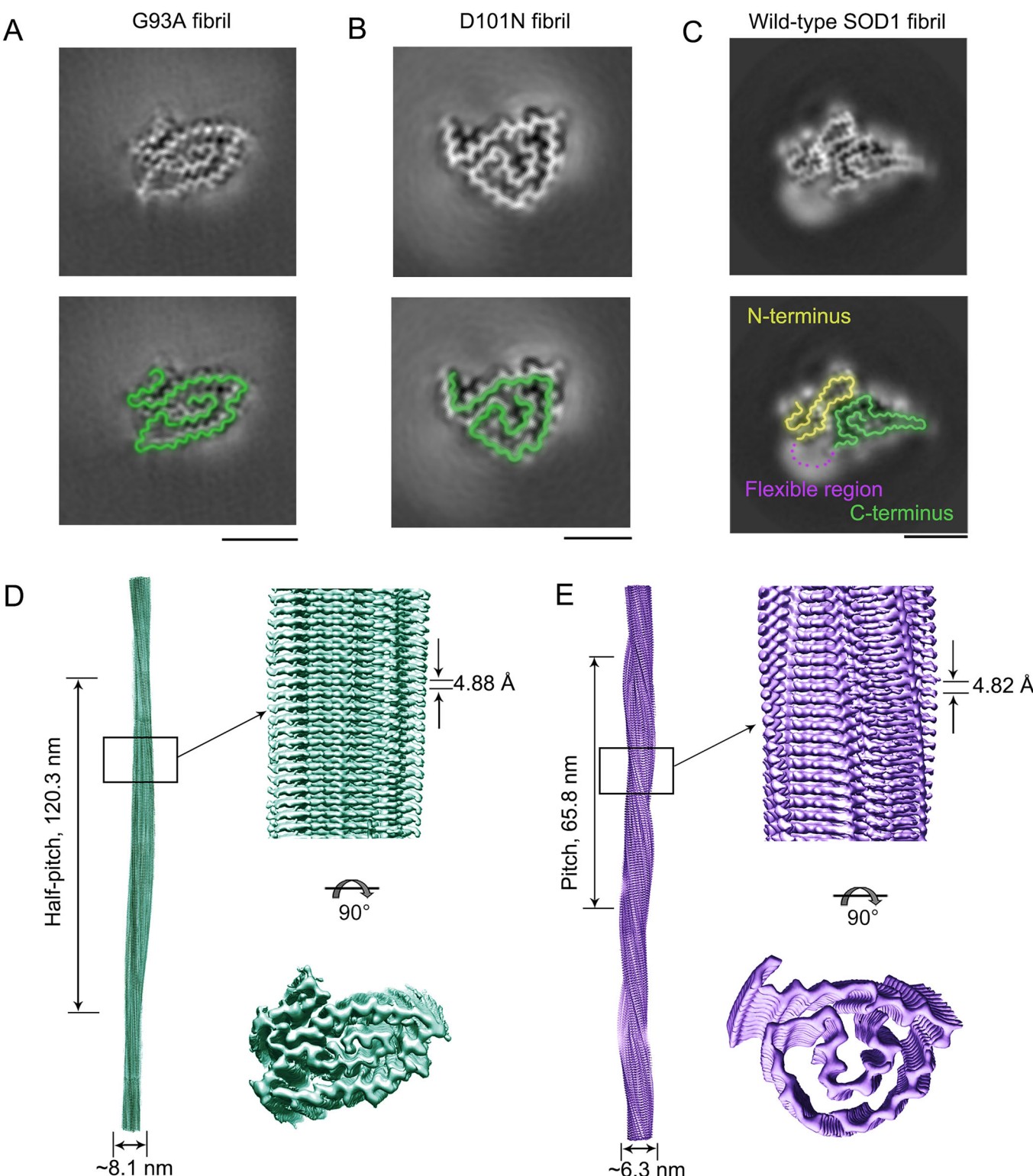

**A** G93A fibril

**B** D101N fibril

**C** Wild-type SOD1 fibril

N-terminus

Flexible region

C-terminus

**D** Half-pitch, 120.3 nm — 4.88 Å — 90° — ~8.1 nm

**E** Pitch, 65.8 nm — 4.82 Å — 90° — ~6.3 nm

Ser107, His110, Glu132, Ser134, Thr137, Asn139, Ser142, Arg143, and Gln153 are located in the interior of the D101N fibril and form a hydrophilic cavity (Fig. 3B).

Importantly, Asp96 and Arg143 form a salt bridge with a distance of 2.7 Å to stabilize the G93A fibril core (Fig. EV5A,B), whereas Asp132 and Arg143 form a salt bridge with a distance of

2.9 Å to stabilize the D101N fibril core (Fig. EV6A,B). Two pairs of amino acids (Asn139 and Arg143 and Gln153 and Arg143) form two hydrogen bonds to stabilize the G93A fibril core (Fig. EV5A,B), whereas three pairs of amino acids (Asn101 and Lys136, Cys111 and Gln153, and Asn139 and Ser105) form four hydrogen bonds to stabilize the D101N fibril core (Fig. EV6A,C–E). The fibril core

◀ **Figure 1. Comparison of the cryo-EM structures of the G93A fibril and the D101N fibril.**

(A, B) Cross-sectional view of the 3D map of the G93A fibril (A) or the D101N fibril (B) showing a protofilament comprising a C-terminal segment (green) and improving by showing the cross-sectional view of one rung. (C) Cross-sectional view of the 3D map of the wild-type SOD1 fibril showing a protofilament comprising not only a C-terminal segment (green) but also an N-terminal segment (yellow) with an unstructured flexible fragment (magenta dashed line), and the 3D map is reused from Fig. 1B (Wang et al, 2022). Scale bars, 5 nm. For full clarity, we false color the equivalent regions in (A–C). (D, E) 3D map of the G93A fibril (D) or the D101N fibril (E) showing a single protofilament (in light green for (D) and light purple for (E)) intertwined into a left-handed helix, with a fibril core width of ~8.1 (D) or ~6.3 nm (E) and a half-helical pitch of 120.3 (D) or a helical pitch of 65.8 nm (E) (left). We have adjusted the pitch range shown in (E) to make sure that the indicated pitch range is accurate. Enlarged section of the G93A fibril (D) or the D101N fibril (E) showing a side view of the density map (top right). Close-up view of the density map on the left showing that the subunit in a protofilament stacks along the fibril axis with a helical rise of 4.88 (D) or 4.82 Å (E) (top right). Top view of the density map of the G93A fibril (D) or the D101N fibril (E) (bottom right).

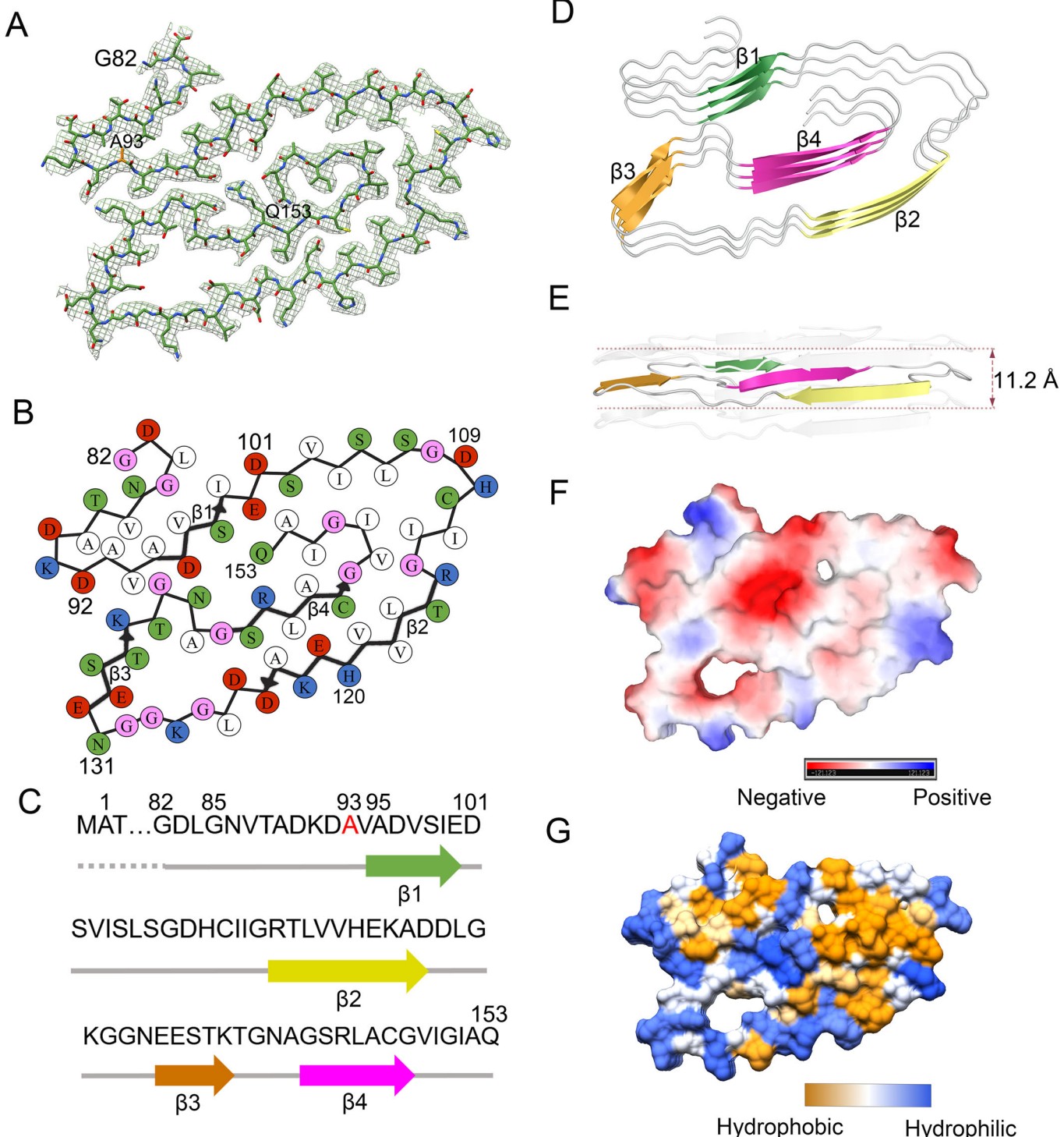

**Figure 2. The ALS-causing SOD1 mutant G93A forms a novel amyloid fibril structure.**

(A) Cryo-EM map of the G93A fibril with the atomic model overlaid. The G93A fibril core comprises a C-terminal segment (residues 82 to 153) colored light green and the ALS-causing mutation site Ala93 is highlighted in orange. (B) Schematic view of the G93A fibril core. The residues are colored as follows: white, hydrophobic; green, polar; red and blue, negatively charged and positively charged, respectively; and magenta, glycine. β strands are indicated with bold lines. (C) Sequence of the G93A fibril core comprising residues 82–153 from the full-length human G93A SOD1 (1 to 153) with the observed four β strands colored green (β1), yellow (β2), orange (β3), and magenta (β4) in the C-terminal segment. The dotted line corresponds to residues 1–81 not modeled in the cryo-EM density. The ALS-causing mutation site Ala93 is highlighted in red. (D) Ribbon representation of the structure of a G93A fibril core containing three molecular layers and a C-terminal segment. (E) As in (D) but viewed perpendicular to the helical axis, revealing that the height of one layer along the helical axis is 11.2 Å. (F) Electrostatic surface representation of the structure of a G93A fibril core containing three molecular layers and a C-terminal segment. (G) Hydrophobic surface representation of the structure of the G93A fibril core shown in (D). The surface of the G93A fibril core is shown according to the electrostatic properties (red, negatively charged; blue, positively charged) (F) or the hydrophobicity (yellow, hydrophobic; blue, hydrophilic) (G) of the residues.

structures of G93A and D101N only comprise a C-terminal segment containing residues 82 to 153 and 95 to 153, respectively (Figs. 2A–D and 3A–D). The wild-type SOD1 fibril core structure, however, comprises not only an N-terminal segment (residues 3 to 55) but also a C-terminal segment (residues 86 to 153), with an unstructured flexible region between them (Wang et al, 2022). Thus, the G93A mutation and the D101N mutation alter crucial salt bridges in wild-type SOD1 fibril and form key salt bridges to stabilize the fibril cores (Figs. EV5 and EV6), resulting in amyloid fibrils with structures distinct from those of wild-type fibril (Fig. 1–3).

The G93A fibril core features a very compact and serpentine fold (Fig. 2A,B) containing four β-strands (Fig. 2C,D) stabilized by two hydrophobic cavities (Fig. 2B,G), whereas the D101N fibril core features a less compact fold and displays a totally different serpentine fold (Fig. 3A,B) containing four β-strands (Fig. 3C,D) stabilized by a hydrophobic cavity (Fig. 3B,G). The four β-strands (β1 to β4) present in the fibril core structures of G93A and D101N are totally different from each other (Figs. 2B–D and 3B–D). The height of one layer of the G93A fibril core (or the D101N fibril core) along the helical axis is 11.2 Å (or 21.1 Å), which is the distance between the highest point in Asp83 (or Ile99) before β1 and the lowest point in Ile149 (or Asn139) after β4 (Figs. 2E and 3E). Together, these results demonstrate that bacterial-purified G93A and D101N mutants form distinct amyloid fibril structures.

## Fibril seeds from bacterial-purified G93A and D101N mutants exhibit different cytotoxicity in neuronal cells

Given that bacterial-purified G93A and D101N do form fibril conformers that differ from each other (Figs. 2 and 3), we predicted that these two mutants might perform distinct functions as those implicated in ALS, exhibiting different toxicity in neuronal cells.

We focused on cytotoxicity to determine the biological relevance of these fibril structures. SH-SY5Y neuroblastoma cells, HEK-293T cells, and HT-22 neuron cells were cultured for 1 day, then 20 mM Tris-HCl buffer (pH 7.4) containing 5 mM TCEP (control), wild-type SOD1 fibril seeds, G93A fibril seeds, or D101N fibril seeds was diluted into tissue culture medium, respectively, and the cells were cultured for 1.5 days and further investigated by an MTT reduction assay and a CCK8 reduction assay (Fig. 4A–F). ELISA assay was used for accurately measuring the concentration of monomers denatured from the SOD1 fibrils (Appendix Fig. S2A), and the final concentration of fibril seeds from bacterial-purified wild-type SOD, G93A, and D101N using ELISA assay was slightly smaller than 10 μM. Because the length of the fibrils may affect the results as well, negative-stain electron

microscopy (NS-EM) images of the input fibrils for G93A (Appendix Fig. S2B), D101N (Appendix Fig. S2C), and wild-type SOD1 (Appendix Fig. S2D) fibrils are shown. Our ELISA assays accurately quantify monomers released from fibrils, and Appendix Fig. S2 shows that all fibrils are similar in length. Abundant short fibrils with similar lengths were observed (Appendix Fig. S2B–D). Notably, fibril seeds from bacterial-purified G93A mutant exhibited significantly higher cytotoxicity to SH-SY5Y cells (Fig. 4A,B), HEK-293T cells (Fig. 4C,D), and HT-22 neuron cells (Fig. 4E,F) than fibril seeds from bacterial-purified wild-type SOD1 ($P = 0.010$, 0.00013, 0.0219, 0.0446, 0.0057, and 0.0063, respectively). Fibril seeds from bacterial-purified D101N mutant, however, did not show significantly greater cytotoxicity to SH-SY5Y cells (Fig. 4A,B), HEK-293T cells (Fig. 4C,D), and HT-22 neuron cells (Fig. 4E,F) than the wild-type SOD1 fibril seeds ($P = 0.349$, 0.0953, 0.523, 0.536, 0.612, and 0.189, respectively). Together, the data showed that G93A fibrils are notably more toxic while D101N fibrils are not significantly more toxic than wild-type SOD1 fibrils.

Together, these data demonstrate that bacterial-purified G93A and D101N form distinct amyloid fibril structures and strongly suggest that these different SOD1 mutants exhibit different toxicity in neuronal cells, among which G93A is the most toxic SOD1 mutant, contributing to ALS pathology.

## Comparison of the images of G93A fibrils and D101N fibrils formed by N-terminally acetylated SOD1 mutants

Human SOD1 is post-translationally modified, removing the initiating Met and acetylating the N-terminus at Ala1. We expand on the limitations of the aforementioned study. First, the recombinant SOD1 mutants we used previously are expressed in *Escherichia coli* where the N-terminus is not acetylated. Second, the aforementioned study does not consider the influence of N-terminal acetylation of SOD1 on the structures and functions of G93A fibrils and D101N fibrils though any covalent difference in the structure can be critical to propensity to form fibrils. Such an influence could be especially pronounced for the case of the A4V ALS-causing mutant, whose mutation site Val4 is at the N-terminus of the protein. Therefore, the impact of our studies would be considerably enhanced by testing our basic conclusions in other experimental systems that contain the usually N-terminally acetylated SOD1 (Figs. 5, 6, and EV2C,D). To investigate this, we expressed and purified recombinant full-length wild-type human SOD1 and its G93A and D101N variants in Expi293F cells where the initiating Met was removed and the N-terminus at Ala1 was acetylated, and identified N-terminal acetylation of the mammalian cell-purified G93A (Fig. 5A), D101N (Fig. 5B), and wild-type SOD1 (Fig. 5C) using mass spectrometry (MS). Analysis of the b-ions in Fig. 5A–C indicated

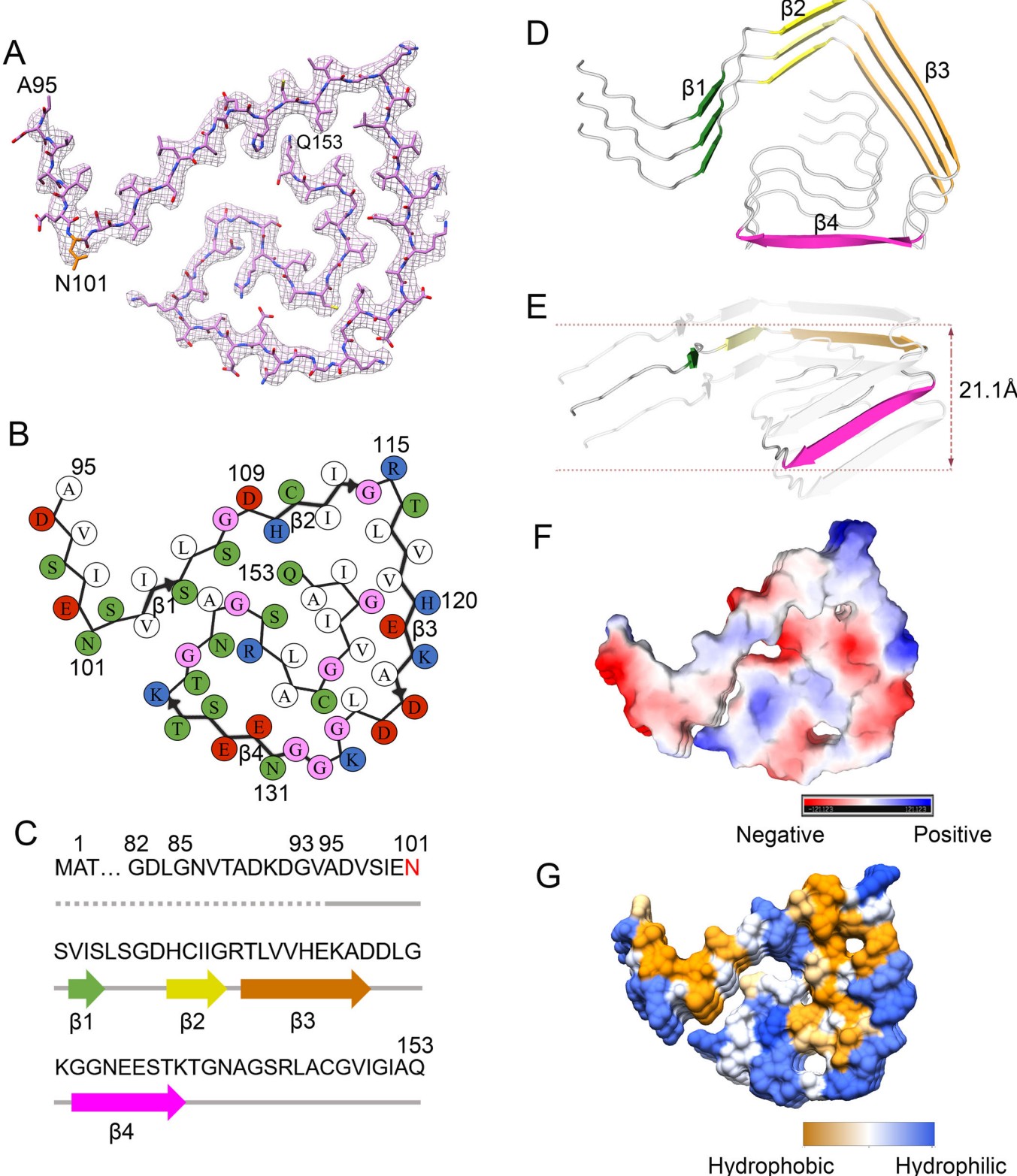

A95

N101

Q153

B

95

109

115

153

101

120

131

C

1    82  85        93 95      101

MAT… GDLGNVTADKDGVADVSIEN

SVISLSGDHCIIGRTLVVHEKADDLG

β1    β2    β3

153

KGGNEESTKTGNAGSRLACGVIGIAQ

β4

D

β1    β2    β3    β4

E

21.1Å

F

Negative        Positive

G

Hydrophobic        Hydrophilic

+42.01037, +42.01014, and +42.01014 Da mass shifts representing the addition of an acetyl group to Ala1, demonstrating N-terminal acetylation in the mammalian cell-purified G93A, D101N, and wild-type SOD1.

We produced amyloid fibrils from recombinant, full-length apo human SOD1 (residues 1–153) with G93A mutation or D101N mutation overexpressed in Expi293F cells by incubating the purified apoproteins in 20 mM Tris-HCl buffer (pH 7.4) containing

◀ **Figure 3. The ALS-causing SOD1 mutant D101N also forms a novel amyloid fibril structure.**

(A) Cryo-EM map of the D101N fibril with the atomic model overlaid. The D101N fibril core comprises a C-terminal segment (residues 95 to 153) colored light purple and the ALS-causing mutation site Asn101 is highlighted in orange. (B) Schematic view of the D101N fibril core. The residues are colored as follows: white, hydrophobic; green, polar; red and blue, negatively charged and positively charged, respectively; and magenta, glycine. β strands are indicated with bold lines. (C) Sequence of the D101N fibril core comprising residues 95–153 from the full-length human D101N SOD1 (1 to 153) with the observed four β strands colored green (β1), yellow (β2), orange (β3), and magenta (β4) in the C-terminal segment. The dotted line corresponds to residues 1–94 not modeled in the cryo-EM density. The ALS-causing mutation site Asn101 is highlighted in red. (D) Ribbon representation of the structure of a D101N fibril core containing three molecular layers and a C-terminal segment. (E) As in (D) but viewed perpendicular to the helical axis, revealing that the height of one layer along the helical axis is 21.1 Å. (F) Electrostatic surface representation of the structure of a D101N fibril core containing three molecular layers and a C-terminal segment. (G) Hydrophobic surface representation of the structure of a D101N fibril core shown in (D). The surface of the D101N fibril core is shown according to the electrostatic properties (red, negatively charged; blue, positively charged) (F) or the hydrophobicity (yellow, hydrophobic; blue, hydrophilic) (G) of the residues. We have reversed the color bar shown in (G) to make sure that the indicated color bar is accurate.

5 mM TCEP and shaking at 37 °C for 34–36 h (see "Methods"). Amyloid fibrils formed by G93A and D101N under these reducing conditions were also concentrated to ~30 μM in a centrifugal filter (Millipore) and examined by electron microscopy without further treatment. Negative-staining TEM images showed that the apo forms of N-terminally acetylated G93A, D101N, and wild-type SOD1 all formed homogeneous and unbranched fibrils under reducing conditions (Fig. 5D–F). The cryo-EM micrographs (Fig. 5G–I) and 2D class average images obtained using RELION3.1 (Scheres, 2020) showed that the G93A fibril and the D101N fibril formed by N-terminally acetylated SOD1 mutants and the wild-type fibril formed by N-terminally acetylated wild-type SOD1 were composed of a single protofilament less intertwined (Fig. EV2C) and intertwined (Fig. EV2D; Appendix Fig. S3), respectively. The 2D class average images showed that at least in vitro, N-terminally acetylated G93A and D101N form fibril conformers that differ from each other and from those formed from bacterial-purified G93A and D101N (Fig. EV2A–D) as well as N-terminally acetylated wild-type SOD1 (Appendix Fig. S3).

Together, these results demonstrate that N-terminally acetylated G93A and D101N mutants also form distinct amyloid fibril structures, confirming our basic conclusion in other experimental systems that contain the usually N-terminally acetylated SOD1.

### Fibril seeds from N-terminally acetylated G93A and D101N are more cytotoxic to cultured cells than are wild-type SOD1 fibril seeds generated under the same conditions

We then investigated the influence of N-terminal acetylation of SOD1 on the cytotoxicity of G93A fibrils and D101N fibrils. SH-SY5Y neuroblastoma cells, HEK-293T cells, and HT-22 neuron cells were cultured for 1 day, then 20 mM Tris-HCl buffer (pH 7.4) containing 5 mM TCEP (control) or fibril seeds from N-terminally acetylated wild-type SOD1, G93A, or D101N was diluted into tissue culture medium, respectively, and the cells were cultured for 1.5 days and further investigated by an MTT reduction assay and a CCK8 reduction assay (Fig. 6A–F). The final concentration of fibril seeds from N-terminally acetylated wild-type SOD, G93A, and D101N using ELISA assay (Appendix Fig. S2A) was also slightly smaller than 10 μM. Because the length of the fibrils may affect the results as well, NS-EM images of the input fibrils for G93A (Appendix Fig. S2E), D101N (Appendix Fig. S2F), and wild-type SOD1 (Appendix Fig. S2G) fibrils, formed by N-terminally acetylated SOD1 proteins, are shown. Abundant short fibrils with similar lengths were also observed (Appendix Fig. S2E–G). Notably,

fibril seeds from N-terminally acetylated G93A mutant also exhibited significantly higher cytotoxicity to SH-SY5Y cells (Fig. 6A,B), HEK-293T cells (Fig. 6C,D), and HT-22 neuron cells (Fig. 6E,F) than fibril seeds from N-terminally acetylated wild-type SOD1 ($P = 0.016$, 0.0081, 0.0477, 0.0126, 0.0026, and 0.0173, respectively). Surprisingly, fibril seeds from N-terminally acetylated D101N mutant showed significantly greater cytotoxicity to SH-SY5Y cells (Fig. 6A,B), HEK-293T cells (Fig. 6C,D), and HT-22 neuron cells (Fig. 6E,F) than the wild-type SOD1 fibril seeds ($P = 0.0143$, 0.0182, 0.0458, 0.0189, 0.00017, and 0.0107, respectively). Together, the data showed that G93A fibrils and D101N fibrils, formed by N-terminally acetylated SOD1 mutants, are notably more toxic than are wild-type SOD1 fibrils generated under the same conditions.

Together, these data demonstrate that N-terminally acetylated G93A and D101N form distinct amyloid fibril structures. Both mutants exhibit surprisingly similar toxicity in neuronal cells, suggesting a possible role of N-terminal acetylation of SOD1 in the cytotoxicity of G93A fibrils and D101N fibrils, contributing to ALS pathology.

## Discussion

Mutations in SOD1 account for approximately 2 to 6% of all ALS cases (Valentine et al, 2005; Shaw and Valentine, 2007; Polymenidou and Cleveland, 2011; Ajroud-Driss and Siddique, 2015; Ayers and Borchelt, 2021; Forsberg et al, 2023). Because familial mutations in SOD1, such as G93A and D101N, are involved in the pathogenesis of ALS, in which SOD1 forms intracellular fibrillar inclusions (Wang et al, 2002; Valentine et al, 2005; Furukawa et al, 2008; Chattopadhyay and Valentine, 2009; Ayers et al, 2016), it is generally thought that these proteinaceous inclusions could be responsible for neuronal cell death in patients with ALS (Valentine et al, 2005; Chattopadhyay and Valentine, 2009; Ayers et al, 2016). Here, we compared the structures of apo SOD1, the wild-type SOD1 fibril, the G93A fibril, and the D101N fibril (Fig. 7A–E). Notably, the SOD1 protein adopts largely distinctive secondary and tertiary structures in four different SOD1 structures (apo SOD1, the wild-type SOD1 fibril, the G93A fibril, and the D101N fibril) (Fig. 7A–D), highlighting the phenotypic diversity of SOD1 in physiological and fibrillar states. The full-length apo human SOD1 monomer contains eight β-strands (β1 to β8), two α-helices (α1 and α2), and a single disulfide bond between Cys57 in α1 and Cys146 in β8 (Strange et al, 2003) (Fig. 7A). Once it folds into its fibrillar form under reducing conditions, the SOD1 subunit undergoes a

**Table 1. Cryo-EM data collection, refinement, and validation statistics.**

| | G93A fibril (EMD-60996, PDB 9IYD) | D101N fibril (EMD-60998, PDB 9IYJ) |
|---|---|---|
| **Data collection and processing** | | |
| Magnification | 105,000 | 105,000 |
| Voltage (kV) | 300 | 300 |
| Camera | Gatan K3 (Krios G4) | Gatan K3 (Krios G4) |
| Frame exposure time (s) | 0.08 | 0.08 |
| Movie frames (n) | 40 | 40 |
| Electron exposure (e⁻/Å²) | 60 | 60 |
| Defocus range (μm) | −2.0 to −1.2 | −2.0 to −1.2 |
| Pixel size (Å) | 0.84 | 0.84 |
| Symmetry imposed | C1 | C1 |
| Box size (pixel) | 320 | 400 |
| Inter-box distance (Å) | 26.9 | 33.6 |
| Micrographs collected (n) | 7333 | 9310 |
| Segments extracted (n) | 1,047,453 | 380,377 |
| Segments after Class2D (n) | 893,277 | 321,421 |
| Segments after Class3D (n) | 57,507 | 104,331 |
| Map resolution (Å) | 3.09 | 2.92 |
| FSC threshold | 0.143 | 0.143 |
| Map resolution range (Å) | 1.8−100 | 1.8−100 |
| **Refinement** | | |
| Initial model used | De novo | De novo |
| Model resolution (Å) | 3.09 | 2.92 |
| FSC threshold | 0.143 | 0.143 |
| Model resolution range (Å) | 3.09 | 2.92 |
| Map sharpening B factor (Å²) | −78.81 | −30.00 |
| Model composition | | |
| Nonhydrogen atoms | 1509 | 1245 |
| Protein residues | 216 | 177 |
| Ligands | 0 | 0 |
| B factors (Å²) | | |
| Protein | 55.36 | 88.64 |
| R.m.s. deviations | | |
| Bond lengths (Å) | 0.008 | 0.011 |
| Bond angles (°) | 0.935 | 0.961 |
| **Validation** | | |
| MolProbity score | 2.57 | 2.34 |
| Clashscore | 15.39 | 15.72 |
| Poor rotamers (%) | 0 | 0 |
| Ramachandran plot | | |
| Favored (%) | 65.71 | 85.96 |

**Table 1.** (continued)

| | G93A fibril (EMD-60996, PDB 9IYD) | D101N fibril (EMD-60998, PDB 9IYJ) |
|---|---|---|
| Allowed (%) | 34.29 | 14.04 |
| Disallowed (%) | 0 | 0 |
| Model versus data (CC) | 0.84 | 0.88 |

complete conformational rearrangement. The wild-type human SOD1 fibril core contains six β-strands (β1 to β6) in its N-terminal segment (residues 3–55) and seven β-strands (β7 to β13) in its C-terminal segment (residues 86–153), exhibiting an in-register intramolecular β-strand architecture (Wang et al, 2022) (Fig. 7A,B). In sharp contrast, the fibril cores of G93A and D101N only comprise a C-terminal segment with residues 82–153 and 95–153, respectively, and the four β-strands (β1 to β4) present in the fibril core structures of G93A and D101N are totally different from each other (Fig. 7A,C,D). In the wild-type SOD1 fibril structure (PDB 7VZF) (Wang et al, 2022), where Gly93 is present in a hydrophobic cavity formed by residues Val94, Val97, Ile99, Ile104, Leu106, Ile113, Leu144, Ala145, Val148, Ile149, Ile151, and Ala152, and Asp101 is involved in a strong salt bridge with His43; in the G93A fibril structure (PDB 9IYD), Ala93 forms a new hydrophobic cavity with residues Leu84, Val87, Ala89, Ala95, Val97, and Ile99 (Fig. 2B); and in the D101N fibril structure (PDB 9IYJ), Asn101 forms two new hydrogen bonds with Lys136 (Fig. EV6A,C). Thus, the G93A mutation and the D101N mutation give rise to amyloid fibrils with distinct structures compared to the wild-type SOD1 fibril (Fig. 7B–E). Bacterial-purified G93A and D101N mutants, without N-terminal acetylation, form distinct amyloid fibril structures with a RMSD of 11.916 Å (57–57 Cα atoms), while the wild-type SOD1 fibril could hardly align with the G93A fibril and the D101N fibril with RMSD of 15.588 Å (70–70 Cα atoms) and 16.365 Å (59–59 Cα atoms), respectively (Fig. 7E). Similarly, N-terminally acetylated G93A and D101N mutants form distinct amyloid fibril structures (Fig. EV2C,D).

This study builds on our previous work (Wang et al, 2022) and presents high-resolution cryo-EM structures of in vitro-generated amyloid fibrils from two familial ALS-linked SOD1 mutants. The recent sudden increase in structural determination of amyloid fibrils purified from patient brain has provided evidence linking amyloid polymorphs to specific diseases (Arseni et al, 2022; Arseni et al, 2023). Given the current absence of structures of SOD1 amyloid fibrils purified from the brains of patients with ALS, our findings provide important structural insights into how SOD1 mutations mediate neuropathogenesis in familial ALS. The disease relevance of these in vitro-prepared amyloid fibrils is addressed as follows. We show that in vitro-generated amyloid fibrils from bacterial-purified G93A mutant, without N-terminal acetylation, are more cytotoxic while those from bacterial-purified D101N mutant are not significantly more cytotoxic than are wild-type SOD1 fibrils generated under the same conditions, which is one of the reasons why G93A SOD1 fibrils have a higher templating capacity in cells of SOD1-G93A transgenic mice (Gurney et al, 1994). It has been reported that the injection of these amyloid

# Bacterial-purified SOD1

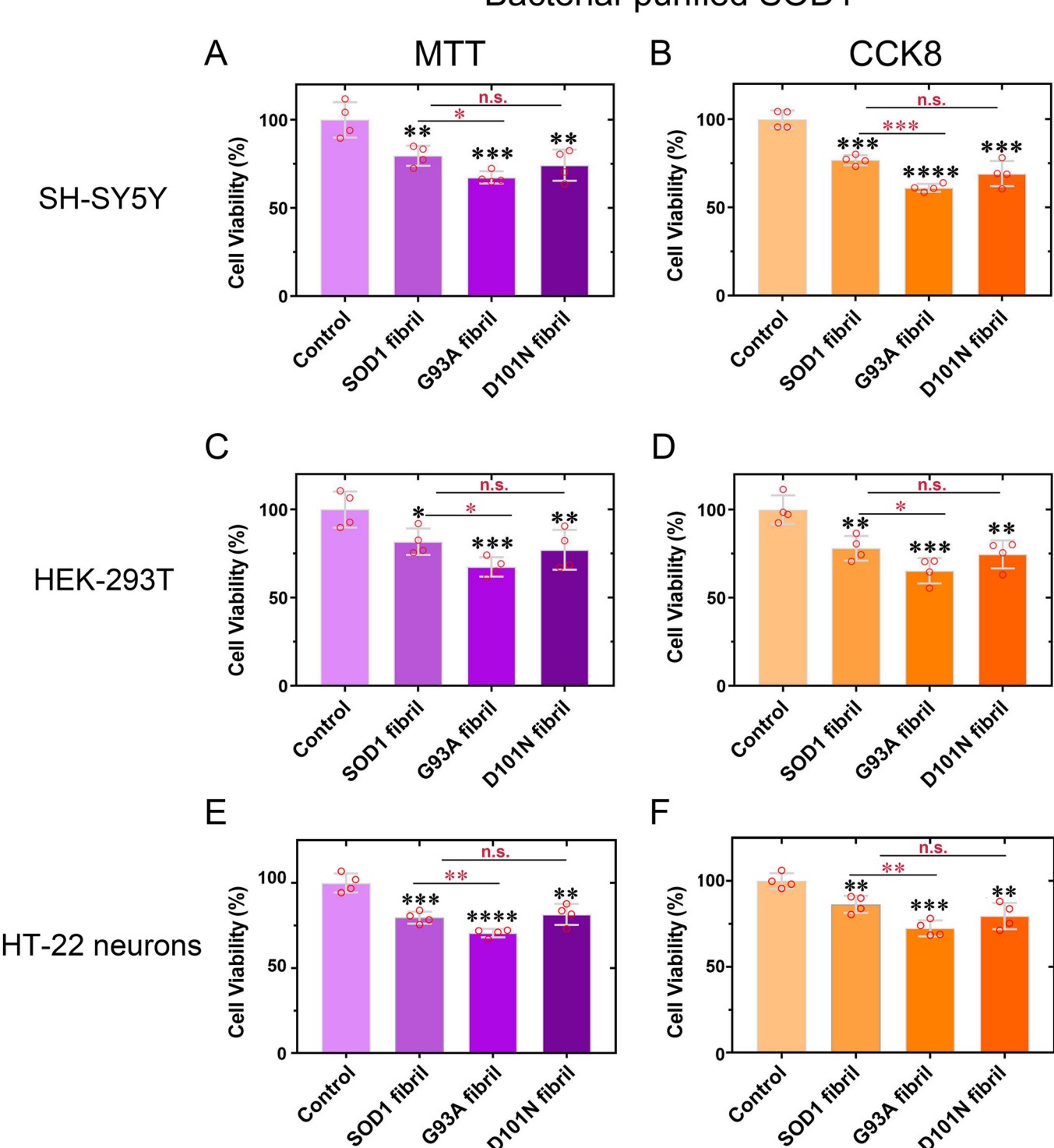

fibrils into the spinal cords of newborn transgenic mice expressing G85R accelerates the progression of ALS (Ayers et al, 2016; Ayers et al, 2021). These amyloid fibrils may be biochemically a forme fruste of those that accumulate in patients with ALS and the same SOD1 mutations with N-terminal acetylation. Interestingly, we show that in vitro-generated amyloid fibrils from N-terminally acetylated G93A and D101N mutants are significantly more toxic than are wild-type SOD1 fibrils generated under the same conditions.

Previous studies proposed two alternative models of the core of G93A fibrils extracted from the spinal cords of transgenic mice carrying this mutation with N-terminal acetylation and developing

◄

**Figure 4. Fibril seeds from bacterial-purified G93A are more cytotoxic to cultured cells than are wild-type SOD1 fibril seeds generated under the same conditions.**

(A–F) Cytotoxicity of fibril seeds from bacterial-purified G93A and D101N to SH-SY5Y neuroblastoma cells (A, B), HEK-293T cells (C, D), or HT-22 neuron cells (E, F) assessed by the MTT assay (A, C, E) and the CCK8 assay (B, D, F) compared with that of fibril seeds from bacterial-purified wild-type SOD1. The cells were cultured for 1 day, then 20 mM Tris-HCl buffer (pH 7.4) containing 0 μM SOD1 fibril seeds, wild-type SOD1 fibril seeds, G93A fibril seeds, or D101N fibril seeds was diluted into tissue culture medium, respectively, and the cells were further cultured for 1.5 days. The final concentration of fibril seeds from bacterial-purified wild-type SOD, G93A, and D101N was slightly smaller than 10 μM. The cell viability (%) (open red circles shown in scatter plots) is expressed as the mean ± SD (with error bars) of values obtained in $n = 4$ (A–F) biologically independent experiments. SOD1 fibrils, $P = 0.0062, 0.00020, 0.0347, 0.0064, 0.00085,$ and $0.0067$ (A–F); G93A fibrils, $P = 0.00010, 0.0000078, 0.00070, 0.00070, 0.000072, 0.00014$ (black) and $0.010, 0.00013, 0.0219, 0.0446, 0.0057, 0.0063$ (red) (A–F); and D101N fibrils, $P = 0.0086, 0.00041, 0.0093, 0.0043, 0.0045, 0.0037$ (black) and $0.349, 0.0953, 0.523, 0.536, 0.612, 0.189$ (red) (A–F). Statistical analyses were performed using two-sided Student's $t$ tests. Values of $P < 0.05$ indicate statistically significant differences. The following notation was used throughout: *$P < 0.05$, **$P < 0.01$, ***$P < 0.001$, and ****$P < 0.0001$ relative to the control. n.s., not significant. Cells treated with 20 mM Tris-HCl buffer (pH 7.4) containing 5 mM TCEP for 1.5 days were used as a control. Source data are available online for this figure.

an ALS-like phenotype based on protease digestion assays, mass spectrometric analysis, and binary epitope mapping assays (Furukawa et al, 2010; Bergh et al, 2015). One "three key region model" predicts that the G93A fibril core contains one N-terminal segment comprising residues 1–30 (Region A), one segment comprising residues 90–120 (Region B), and one C-terminal segment comprising residues 135–153 (Region C) (Furukawa et al, 2010). This is partly compatible with our model, wherein β1 and β2 correspond to Region B and β3 and β4 correspond to Region C in the model of the G93A fibril core proposed by Furukawa and co-workers (Furukawa et al, 2010). Another "three key region model" predicts that the G93A fibril core contains two N-terminal segments comprising residues 43–57 and 57–72 and one C-terminal segment comprising residues 131–153 (Bergh et al, 2015). This finding is compatible with our model to a lesser extent, wherein β3 and β4 perfectly correspond to the C-terminal segment in the model of the G93A fibril core proposed by Bergh and co-workers (Bergh et al, 2015). It should be mentioned that in our model, the fibril cores of G93A and D101N contain the C-terminal segment comprising residues 82–153 and 95–153, respectively. The N-terminal segments in the fibril cores were not observed. This differs from the findings of Furukawa et al (2010) and Bergh et al (2015), who reported that one or two N-terminal segments are present in the G93A fibril core. Because it raised concern that the protein was degraded since only C-terminal protein was incorporated in the ordered core of structures, the SDS–PAGE gels of SOD1 protein before and after in vitro aggregation have been added to show that full-length SOD1 was incorporated into the SOD1 fibrils. It should be pointed out that the structures are limited to bacterially expressed SOD1 proteins and the heart of the paper is the cryo-EM structures of in vitro fibrils formed by bacterial-purified SOD1 mutants G93A and D101N. The following are the limitations using bacterial-purified proteins as well as the divergence from amyloid fibril structures formed by ALS-causing SOD1 mutants and the "three key region model". First, we considered the impact of cofactors that might be present in patients and mouse models of ALS but not in our cell-free fibrillization studies. Notably, such cofactors might promote the inclusion of more N-terminal residues in the fibril core, as suggested for these mutants by Furukawa et al (2010) and Bergh et al (2015) and as are present in the wild-type SOD1 fibril core (Wang et al, 2022). Second, we considered the impact of N-terminally acetylated SOD1 mutants that are present in patients and mouse models of ALS but not in our cell-free fibrillization studies. We suggest that N-terminal acetylation of SOD1 mutants

might promote the inclusion of more N-terminal residues in the fibril core. Although the SOD1 fibril structures determined here may not be identical to those that accumulate in patients with ALS and the same SOD1 mutations that are N-terminally acetylated, this work provides important initial insights into the potential structural underpinnings of how these SOD1 mutants might aggregate and cause cytotoxicity in ALS. We plan to determine cryo-EM structures of in vitro-generated amyloid fibrils from N-terminally acetylated G93A and D101N mutants and collect structural data on the G93A fibril and the D101N fibril purified from the brains of patients with ALS or ALS transgenic mice in the near future.

Our results hint that fibrils formed by mammalian-expressed, N-terminally acetylated SOD1 mutants adopt completely different conformations from those made by SOD1 mutants purified from bacteria. We then discuss how these structural differences relate to cytotoxicity. We demonstrate that bacterial-purified G93A and D101N, without N-terminal acetylation, exhibit different toxicity in neuronal cells, among which G93A is the most toxic SOD1 mutant. Mammalian-expressed, N-terminally acetylated G93A and D101N, however, exhibit similar toxicity in neuronal cells, suggesting possible roles of these structural differences and N-terminal acetylation of SOD1 in the cytotoxicity of G93A fibrils and D101N fibrils.

Both G93A fibrils and D101N fibrils analyzed in this study were generated under reducing conditions, resulting in structures lacking copper and zinc and a disulfide bridge. These immature, metal-free, and disulfide-reduced forms are hypothesized to represent the cytotoxic misfolded SOD1 conformations responsible for pathogenesis (Tiwari and Hayward, 2003; Tiwari et al, 2005; Furukawa et al, 2008; Luchinat and Banci, 2018; Sala et al, 2019). Supporting this, amyloid-like aggregates extracted from ALS transgenic mice expressing disease-causing SOD1 mutants like G93A consist predominantly of metal-deficient and disulfide-reduced SOD1 (Jonsson et al, 2006; Zetterström et al, 2007), indicating their disease-causing potential. Reinforcing this finding, recent work by Hale and colleagues demonstrated that metal-deficient SOD1 species are prevalent in spinal cord and brain tissues of SOD1-G93A transgenic mice (Hale et al, 2024). Furthermore, disulfide reduction susceptibility has been identified in over ten ALS-associated SOD1 mutants (Tiwari and Hayward, 2003). Notably, four of these mutants, including G93A, are consistently enriched in the spinal cord and brain of transgenic mice from early to late disease stages and lack native disulfide bonds (Jonsson et al, 2006; Zetterström et al, 2007). Consequently, determining the

atomic structures of mutant SOD1 fibrils is essential for elucidating the pathogenic mechanisms of SOD1 mutants in ALS.

In summary, on the one hand, two SOD1 metal-binding mutants, G93A and D101N, which have prion-like properties (Ayers et al, 2014a; Ayers et al, 2021; Ayers et al, 2023) and were previously found to induce ALS-like disease in a mouse model (Gurney et al, 1994; Cleveland and Rothstein, 2001; Olsen et al, 2001; Jonsson et al, 2006; Ayers et al, 2014b; Zhang et al, 2014; Bergh et al, 2015; Ayers et al, 2021; Liu et al, 2024), form fibril conformers that differ from each other and from those formed from wild-type SOD1, as revealed by cryo-EM; the G93A fibril and the D101N fibril consist of a single protofilament with a fibril core comprising residues 82–153 and 95–153, respectively. The reported high-resolution cryo-EM structures of the G93A fibril and the D101N fibril reveal unusual overall structures compared to that of the wild-type fibril, characterized by alterations in crucial salt bridges, a C-terminal fibril core, four instead of thirteen β strands in the cores, two hydrophilic cavities and a hydrophobic cavity, and two and four hydrogen bonds, respectively. On the other hand, fibril seeds from bacterial-purified G93A are significantly more toxic than fibril seeds from bacterial-purified D101N, which do not show a marked increase in toxicity compared to wild-type fibril seeds, and G93A is the most toxic SOD1 mutant among them. The fibril structures will be valuable for understanding the structural basis underlying the functions of familial mutations in the amyloid state and inspiring future research on the molecular mechanisms by which mutations in SOD1 exhibit cytotoxicity in neuronal cells and cause the familial form of ALS.

# Methods

### Reagents and tools table

| Reagent/resource | Reference or source | Identifier or catalog number |
|---|---|---|
| **Experimental models** | | |
| 2003Expi293F™ cells | Gibco™ | Cat. No. A14527 |
| SH-SY5Y neuroblastoma cells | China Center for Type Culture Collection | Cat. No. GDC0210 |
| HEK-293T cells | China Center for Type Culture Collection | Cat. No. GDC0187 |
| HT-22 neuron cells | China Center for Type Culture Collection | Cat. No. GDC0673 |
| **Recombinant DNA** | | |
| pET-22b SOD1 | Thomas O'Halloran Lab | Cat. No. 69744-3 |
| pET-22b SOD1 G93A | Site mutation | Cat. No. 69744-3 |
| pET-22b SOD1 D101N | Site mutation | Cat. No. 69744-3 |
| pcDNA3.4 SOD1 | Gibco™ | Cat. No. PL1012 |
| pcDNA3.4 SOD1 G93A | Site mutation | Cat. No. PL1012 |
| pcDNA3.4 SOD1 D101N | Site mutation | Cat. No. PL1012 |
| **Antibodies** | | |

| Reagent/resource | Reference or source | Identifier or catalog number |
|---|---|---|
| **Oligonucleotides and other sequence-based reagents** | | |
| **Chemicals, enzymes and other reagents** | | |
| MTT Cell Proliferation and Cytotoxicity Assay Kit | Beyotime | Cat. No. C0009S |
| Cell Counting Kit-8 | Beyotime | Cat. No. C0037 |
| expifectamine 293 transfection kit | Gibco™ | Cat. No. A14525 |
| superoxide dismutase 1 Elisa Kit | JILID | Custom made |
| Tris(2-carboxyethyl) phosphine | Sigma | Cat. No. 646547 |
| **Software** | | |
| Relion 3.1 | https://relion.readthedocs.io/en/release-3.1/index.html | |
| PyMol 2.3 | https://pymol.org/ | |
| PHENIX 1.15.2 | http://phenix-online.org/download/ | |
| WinCoot | MRC Laboratory of Molecular Biology | |
| Chimera 1.15 | https://www.cgl.ucsf.edu/chimera/ | |
| **Other** | | |

## Protein expression and purification

A plasmid encoding full-length human SOD1 (1–153) was a gift from Dr. Thomas O'Halloran (Chemistry of Life Processed Institute, Northwestern University). The sequence for SOD1 1–153 was expressed from the vector pET-3d, and two SOD1 mutants, G93A and D101N, were constructed by site-directed mutagenesis using a wild-type SOD1 template; the primers used are shown in Appendix Table S1. All the SOD1 plasmids were transformed into *E. coli* BL21 (DE3) cells (Novagen, Merck, Darmstadt, Germany). Recombinant full-length wild-type human SOD1 and its G93A and D101N variants were expressed from the vector pET-22b (+) in *E. coli* BL21 (DE3) cells. SOD1 proteins were purified to homogeneity by Q-Sepharose chromatography as described by Chattopadhyay et al (2008) and Xu et al (2018). The target genes for wild-type SOD1 and its mutants G93A and D101N were subcloned into the pcDNA3.4, and the SOD1 plasmids were transfected into Expi293F cells using PEI max transfection reagent (Polysciences 24765-100). After 4 days of transfection, the cells were harvested by centrifugation at 8000 rpm at 4 °C for 6 min and then washed with $1 \times$ PBS (pH 7.4) for three times. The cells were resuspended with $1 \times$ PBS (pH 7.4) and sonicated at 30 W for 15 min. The supernatant was collected after centrifugation at $10,000 \times g$ for 15 min, then wild-type SOD1 and its G93A and D101N variants were captured by Ni Smart Beads and eluted in imidazole. After purification, recombinant wild-type SOD1 and its G93A and D101N variants were demetallated by dialysis in 10 mM EDTA and 10 mM NaAc buffer (pH 3.8) five times as described by Chattopadhyay et al (2008) and Xu et al (2018). In all, 10 mM NaAc buffer (pH 3.8) and 20 mM Tris-HCl buffer (pH 7.4) were used for

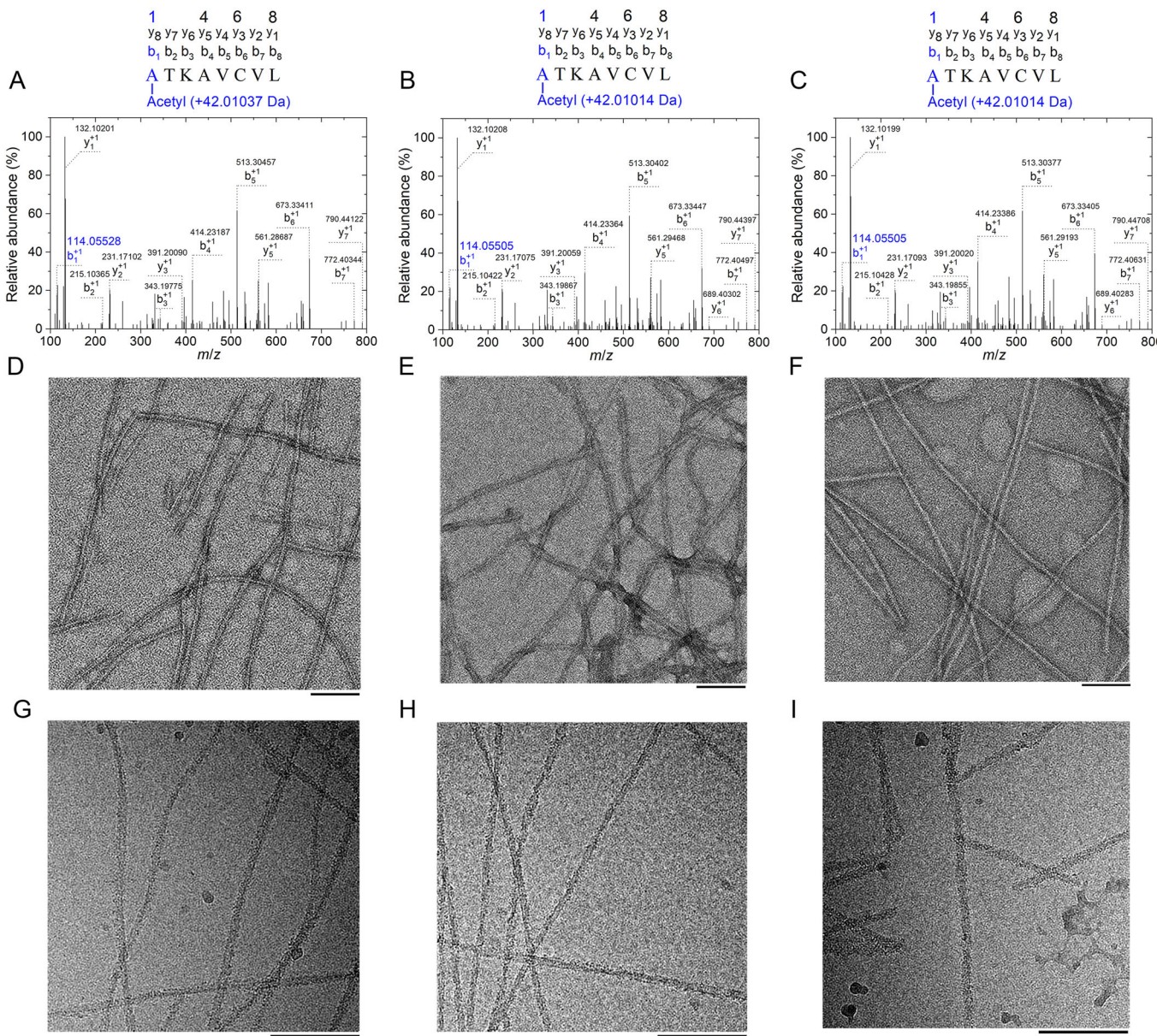

**Figure 5. Comparison of the images of G93A fibrils and D101N fibrils formed by N-terminally acetylated SOD1 mutants.**

Recombinant full-length wild-type human SOD1 and its G93A and D101N variants were expressed and purified in Expi293F cells where the initiating Met was removed and the N-terminus at Ala1 was acetylated. (A–C) Identification of N-terminal acetylation of the mammalian cell-purified G93A (A), D101N (B), and wild-type SOD1 (C) using mass spectrometry (MS). The Coomassie Blue-stained gels of SDS–PAGE of N-terminally acetylated wild-type SOD1 and its G93A and D101N variants were scissored out, chopped, trypsinized, and then analyzed with nano-LC-MS/MS. A daughter ion (MS²) spectrum of parent peptide A¹TKAVCVL⁸ digested by trypsin. (A) Analysis of the b-ion (b₁⁺¹) indicates +42.01037 Da mass shift (114.05528 − 1.0078 = 113.04748 Da, 113.04748 − 71.03711 = 42.01037 Da) representing the addition of an acetyl group to Ala1, demonstrating N-terminal acetylation in the mammalian cell-purified G93A. (B and C) Analysis of the b-ion (b₁⁺¹) indicates +42.01014 Da mass shift (114.05505 − 1.0078 = 113.04725 Da, 113.04748 − 71.03711 = 42.01014 Da) representing the addition of an acetyl group to Ala1, demonstrating N-terminal acetylation in the mammalian cell-purified D101N (B) and wild-type SOD1 (C). (D–F) Negative-staining TEM images of amyloid fibrils produced from N-terminally acetylated G93A (D), D101N (E), and wild-type SOD1 (F). (G–I) Raw cryo-EM images of amyloid fibrils assembled from N-terminally acetylated G93A (G), D101N (H), and wild-type SOD1 (I). The scale bars represent 200 nm (D–F) and 100 nm (G–I), respectively.

further dialysis. The apo forms of wild-type SOD1, G93A, and D101N were then concentrated, filtered, and stored at −80 °C. An AAnalyst-800 atomic absorption spectrometer (PerkinElmer) was used to quantify the metal content of the SOD1 samples. Samples of wild-type SOD1, G93A, and D101N contained <5% residual metal ions, indicating that the samples were indeed in the apo state. SDS − PAGE

and mass spectrometry were used to confirm that the purified apo SOD1 proteins were single species with an intact disulfide bond. A NanoDrop OneC Microvolume UV–Vis Spectrophotometer (Thermo Fisher Scientific) was used to determine the concentration of apo SOD1 proteins according to their absorbance at 214 nm with a standard calibration curve drawn from BSA.

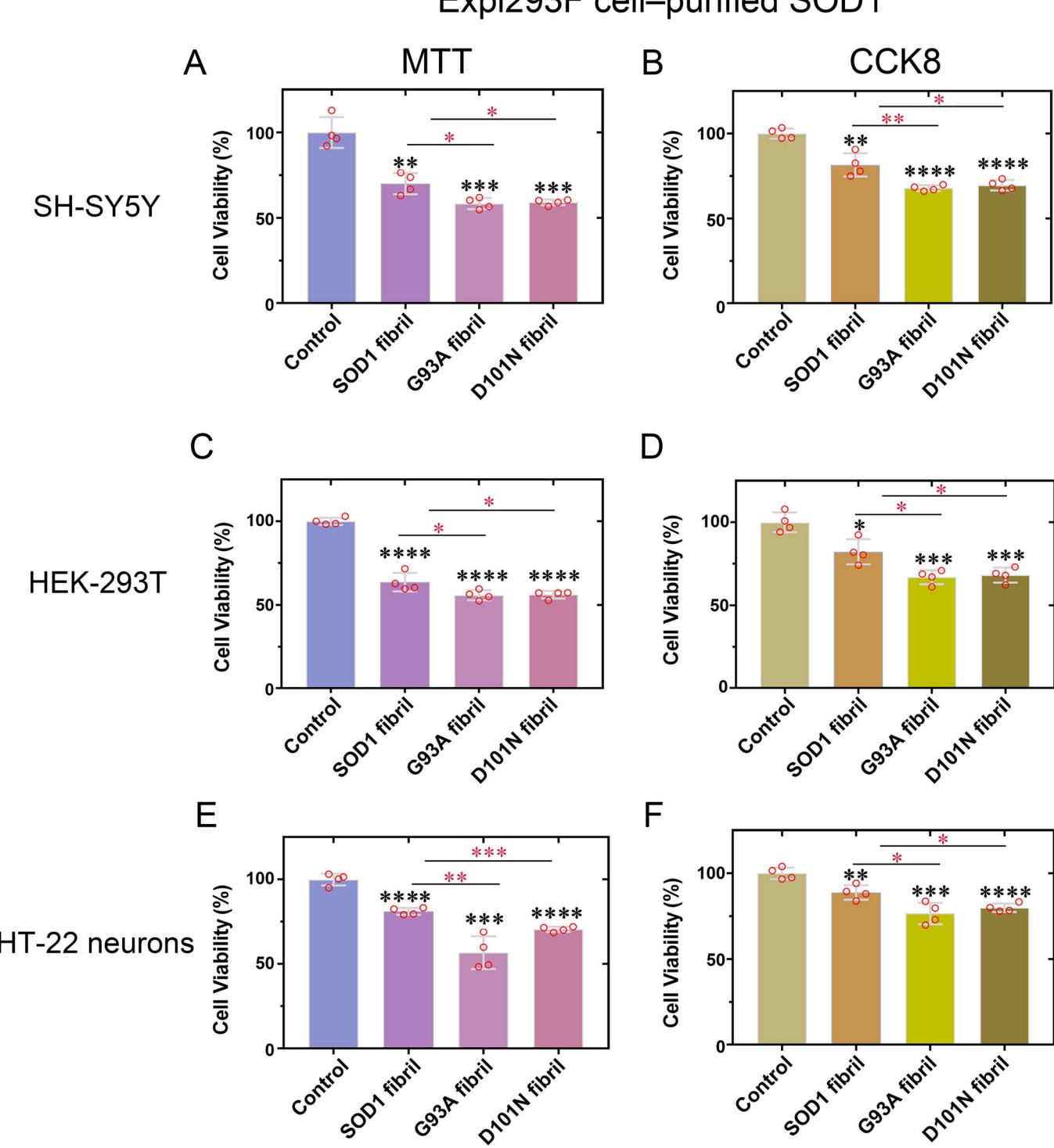

## LC-MS/MS analysis

Full-length wild-type SOD1 and its G93A and D101N variants were expressed and purified in Expi293F cells. The SOD1 proteins were separated by SDS–PAGE. The gel bands of interest were excised from the gel, reduced with 5 mM DTT, and alkylated with 11 mM iodoacetamide, followed by in-gel digestion with sequencing-grade modified trypsin at 37 °C overnight. The peptides were extracted twice with 0.1% trifluoroacetic acid in 50% acetonitrile aqueous solution for 30 min and then dried in a SpeedVac (Eppendorf). Peptides were redissolved in 20 μl of 0.1% trifluoroacetic acid and 6 μl of extracted peptides were analyzed by Orbitrap Exploris 480 mass spectrometer (Thermo Fisher Scientific). For LC-MS/MS analysis, the peptides were separated by a 120-min gradient elution

**Figure 6. Fibril seeds from N-terminally acetylated G93A and D101N are more cytotoxic to cultured cells than are wild-type SOD1 fibril seeds generated under the same conditions.**

(A–F) Cytotoxicity of fibril seeds from Expi293F cell-purified G93A and D101N to SH-SY5Y neuroblastoma cells (A, B), HEK-293T cells (C, D), or HT-22 neuron cells (E, F) assessed by the MTT assay (A, C, E) and the CCK8 assay (B, D, F) compared with that of fibril seeds from Expi293F cell-purified wild-type SOD1. The cells were cultured for 1 day, then 20 mM Tris-HCl buffer (pH 7.4) containing 0 μM SOD1 fibril seeds, wild-type SOD1 fibril seeds, G93A fibril seeds, or D101N fibril seeds was diluted into tissue culture medium, respectively, and the cells were further cultured for 1.5 days. The final concentration of fibril seeds from the mammalian cell-purified wild-type SOD, G93A, and D101N was slightly smaller than 10 μM. The cell viability (%) (open red circles shown in scatter plots) is expressed as the mean ± SD (with error bars) of values obtained in $n = 4$ (A–F) biologically independent experiments. SOD1 fibrils, $P = 0.0016, 0.0027, 0.000017, 0.0104, 0.000091$, and $0.0065$ (A–F); G93A fibrils, $P = 0.00013, 0.0000016, 0.00000036, 0.00010, 0.00016, 0.00061$ (black) and $0.016, 0.0081, 0.0477, 0.0126, 0.0026, 0.0173$ (red) (A–F); and D101N fibrils, $P = 0.00011, 0.0000076, 0.00000015, 0.00014, 0.0000050, 0.000079$ (black) and $0.0143, 0.0182, 0.0458, 0.0189, 0.00017, 0.0107$ (red) (A–F). Statistical analyses were performed using two-sided Student's *t* tests. Values of $P < 0.05$ indicate statistically significant differences. The following notation was used throughout: *$P < 0.05$, **$P < 0.01$, ***$P < 0.001$, and ****$P < 0.0001$ relative to the control. Cells treated with 20 mM Tris-HCl buffer (pH 7.4) containing 5 mM TCEP for 1.5 days were used as a control. Source data are available online for this figure.

at a flow rate 0.30 μl/min with a Thermo-Dionex Ultimate 3000 HPLC system (Thermo Fisher Scientific), which was directly interfaced with an Orbitrap Exploris 480 mass spectrometer (Thermo Fisher Scientific). The analytical column was a home-made fused silica capillary column (75 μm ID, 350 mm length) packed with C-18 resin (1.9 μm, Dr. Maisch GmbH). Mobile phase consisted of 0.1% formic acid, and mobile phase B consisted of 80% acetonitrile and 0.1% formic acid. Orbitrap Exploris 480 mass spectrometer was operated in the data-dependent acquisition mode using Xcalibur 4.5.445.18 software and there was a single full-scan mass spectrum in the orbitrap (350–1600 *m/z*, 60,000 resolution) followed by 2-s data-dependent MS/MS scans in an Ion Routing Multipole at 30 normalized collision energy (HCD). The MS/MS spectra from each LC-MS/MS run were searched against the SOD1.fasta using an in-house Proteome Discoverer (Version PD2.5, Thermo Fisher Scientific). The search criteria were as follows: chymotrypsin was chosen as specific enzyme; two missed cleavage was allowed; carbamidomethylation (C) were set as the fixed modifications; the oxidation (M) and phosphorylation (STY) were set as the variable modification; acetylation was set as dynamic modification in protein N-terminus. Precursor ion mass tolerances were set at 20 ppm for all MS acquired in an orbitrap mass analyzer; and the fragment ion mass tolerance was set at 0.02 Da for all MS2 spectra. Confidence levels were set to 1% FDR (high confidence).

## SOD1 fibril formation

The apo forms of bacterial-purified full-length wild-type human SOD1 and its G93A and D101N variants were incubated in 20 mM Tris-HCl buffer (pH 7.4) containing 5 mM TCEP and shaken at 37 °C for 40–48 h, after which the SOD1 fibrils were collected. The apo forms of Expi293F cell-purified full-length wild-type human SOD1 and its G93A and D101N variants were incubated in 20 mM Tris-HCl buffer (pH 7.4) containing 5 mM TCEP and shaken at 37 °C for 34–36 h, after which the SOD1 fibrils were collected. Large amorphous aggregates in SOD1 fibril samples were removed by centrifugation at $5000 \times g$ at 4 °C for 10 min. The supernatants (purified amyloid fibrils of SOD1) were then concentrated to ~30 μM in a centrifugal filter (Millipore). SDS–PAGE and mass spectrometry were used to confirm that full-length SOD1 was incorporated into the SOD1 fibrils. We sonicated the SOD1 fibrils for 5 min (5 s on, 5 s off) on ice before the cytotoxicity tests so that all fibrils are similar in length. A NanoDrop OneC Microvolume

UV–Vis Spectrophotometer (Thermo Fisher Scientific) was used to determine the concentrations of wild-type SOD1 fibrils, G93A fibrils, and D101N fibrils according to their absorbance at 214 nm with a standard calibration curve drawn from BSA. The samples of G93A fibrils, D101N fibrils, and wild-type SOD1 fibrils were dissolved in 8 M urea and then ELISA assay, using the superoxide dismutase 1 ELISA Kit with antibody-coated plate (JILID, Wuhan), was used for accurately measuring the concentration of monomers denatured from the SOD1 fibrils.

## Coomassie Blue staining

The samples of G93A dimers, D101N dimers, G93A fibrils, and D101N fibrils were dissolved in 8 M urea for more than 2 h. The samples were then boiled in SDS–PAGE loading buffer for 15 min and separated by 12.5% SDS–PAGE. The gels were stained with Coomassie Blue staining solution for 2 h and then washed with destaining buffer until the band of SOD1 protein appeared.

## TEM of G93A fibrils and D101N fibrils

G93A fibrils and D101N fibrils assembled from bacterial-purified G93A and D101N or N-terminally acetylated G93A and D101N were examined by TEM of negatively stained samples. Ten microliters of SOD1 mutation fibril samples (~30 μM) were loaded on copper grids for 30 s and washed with $H_2O$ for 10 s. Samples on grids were then stained with 2% (w/v) uranyl acetate for 30 s and air-dried at 25 °C. The stained samples were examined using a Talos L120C transmission electron microscope (Thermo Fisher Scientific) operating at 120 kV for G93A fibrils and D101N fibrils.

## AFM of G93A fibrils and D101N fibrils

G93A fibrils and D101N fibrils assembled from bacterial-purified G93A and D101N were produced as described above. Ten microliters of SOD1 mutation fibril samples (~30 μM) were incubated on a freshly cleaved mica surface for 2 min, followed by rinsing three times with 10 μl of pure water to remove the unbound fibrils and drying at room temperature. The fibrils on the mica surface were probed in air by a Dimension Icon scanning probe microscope (Bruker) in ScanAsyst mode. The measurements were performed by using a SCANASYST-AIR probe (Bruker) with a spring constant of 0.4 N/m and a resonance frequency of 70 kHz. AFM images with a fixed resolution ($256 \times 256$ data points) were

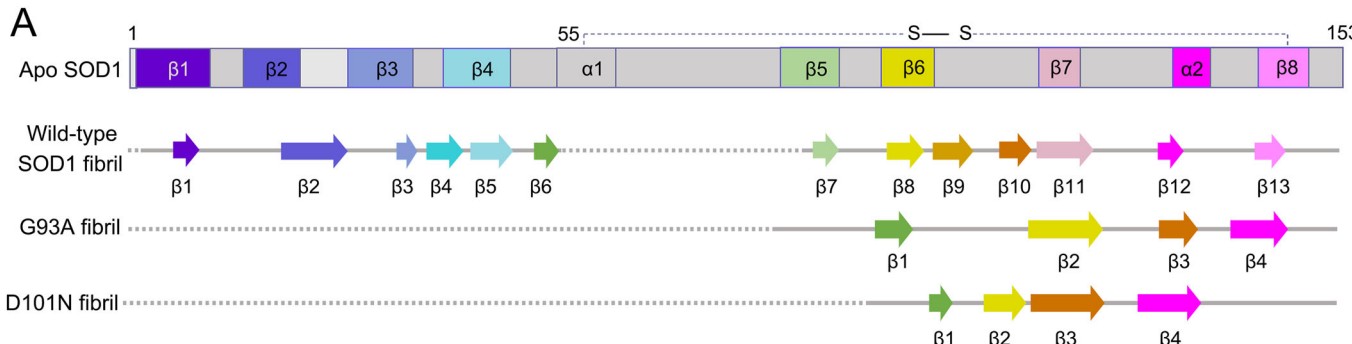

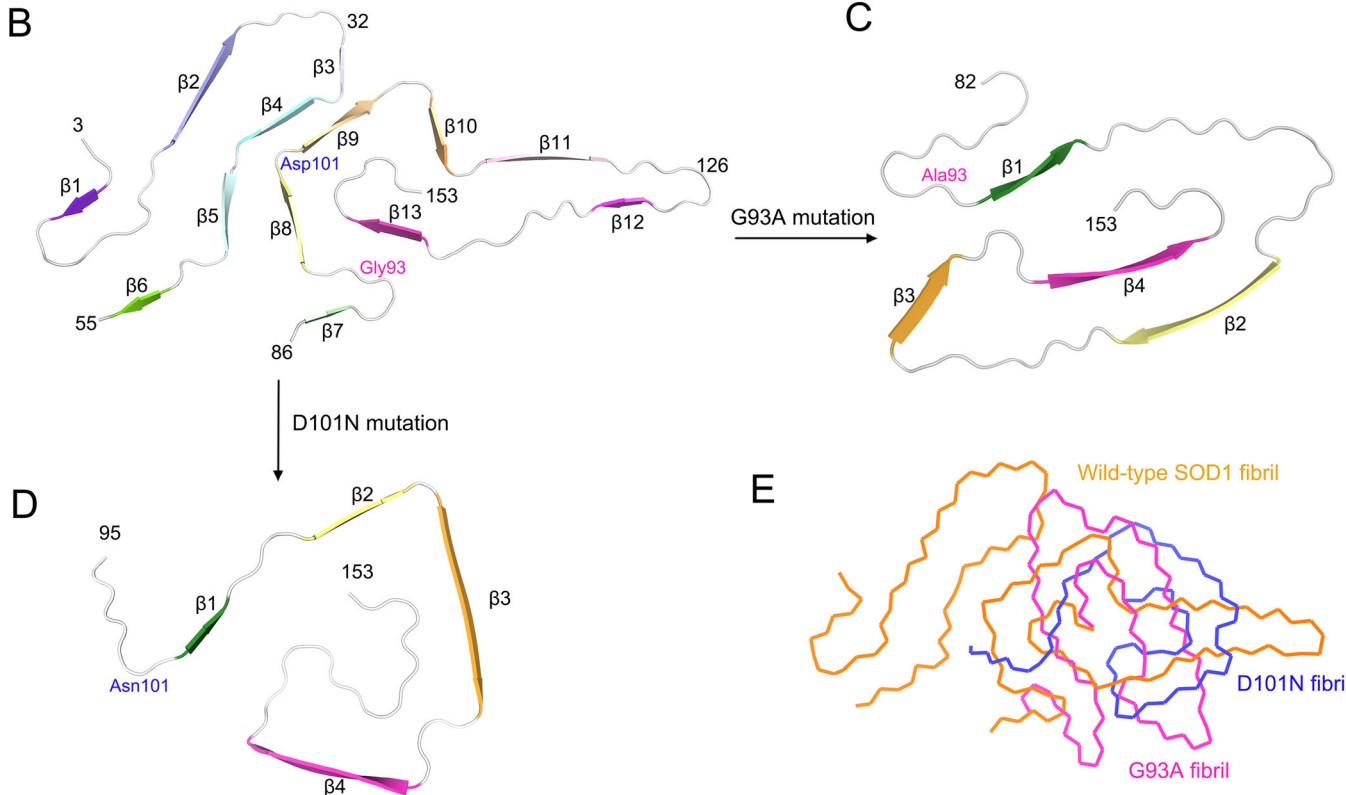

**Figure 7. Comparison of the structures of the apo form of SOD1, the wild-type SOD1 fibril, the G93A fibril, and the D101N fibril.**

(A) Sequence alignment of the full-length apo human SOD1 monomer (PDB 1HL4) (Strange et al, 2003) with eight β-strands (β1 to β8), two α-helices, and a single disulfide bond. Sequence alignment of the wild-type SOD1 fibril core comprising residues 3–55 and 86–153 from the full-length wild-type human SOD1 (PDB 7VZF) (Wang et al, 2022) with the observed thirteen β strands (β1 to β13). Sequence alignment of the G93A fibril core comprising residues 82–153 from the full-length human G93A SOD1 with the observed four β strands colored green (β1), yellow (β2), orange (β3), and magenta (β4) in the C-terminal segment. The dotted line corresponds to residues 1–81 not modeled in the cryo-EM density. Sequence alignment of the D101N fibril core comprising residues 95–153 from the full-length human D101N SOD1 with the observed four β strands colored green (β1), yellow (β2), orange (β3), and magenta (β4) in the C-terminal segment (bottom). The dotted line corresponds to residues 1–94 not modeled in the cryo-EM density. (B–D) Ribbon representation of the structures of a wild-type SOD1 fibril core (B), a G93A fibril core (C), and a D101N fibril core (D), all of which contain one molecular layer and a monomer. Positions of G93A (magenta) and D101N (blue) are labelled in (B–D). (E) Overlay of the structures of a wild-type SOD1 fibril core (orange), a G93A fibril core (magenta), and a D101N fibril core (blue).

acquired at a scan rate of 1 Hz and analyzed by using NanoScope Analysis 2.0 software (Bruker).

## Cryo-EM of G93A fibrils and D101N fibrils

G93A fibrils and D101N fibrils assembled from bacterial-purified G93A and D101N or N-terminally acetylated G93A and D101N

were produced as described above. An aliquot of 3.5 μl of ~30 μM SOD1 mutation fibril solution was applied to glow-discharged holey carbon grids (Quantifoil Cu R1.2/1.3, 300 mesh), blotted for 3.5 s, and plunge-frozen in liquid ethane using a Vitrobot Mark IV. The grids were examined using a Glacios transmission electron microscope operated at 200 kV and equipped with a field emission gun and a Ceta-D CMOS camera (Thermo Fisher Scientific). The

cryo-EM micrographs were acquired on a Krios G4 transmission electron microscope operated at 300 kV (Thermo Fisher Scientific) and equipped with a Bio-Quantum K3 direct electron detector (Gatan). A total of 7333 movies for G93A fibrils assembled from bacterial-purified G93A and 9310 movies for D101N fibrils assembled from bacterial-purified D101N were collected in super-resolution mode at a nominal magnification of $\times 105,000$ (physical pixel size, 0.84 Å) and a dose of 18.75 $e^-$ Å$^{-2}$ s$^{-1}$ (see Table 1). A total of 7799 movies for G93A fibrils assembled from N-terminally acetylated G93A and 5442 movies for D101N fibrils assembled from N-terminally acetylated D101N were collected in super-resolution mode at a nominal magnification of $\times 105,000$ (physical pixel size, 0.824 Å) and a dose of 15.717 $e^-$ Å$^{-2}$ s$^{-1}$. An exposure time of 3.2 s was used, and the resulting videos were dose-fractionated into 40 frames. A defocus range of $-1.2$ to $-2.0$ μm was used.

## Helical reconstruction

All image-processing steps, which included manual picking, particle extraction, 2D and 3D classifications, 3D refinement, and post-processing, were performed by RELION-3.1 (Scheres, 2020). For the G93A fibril assembled from bacterial-purified G93A, 41,408 fibrils were picked manually from 7333 micrographs, and 1024- and 686-pixel boxes were used to extract particles by a 90% overlap scheme. Two-dimensional classification of 1024-box size particles was used to calculate the initial twist angle. With regard to the helical rise, 4.8 Å was used as the initial value. The particles were extracted into 400-box sizes for further processing. After several iterations of 2D and 3D classifications, particles with the same morphology were picked out. Local searches of symmetry in 3D classification were used to determine the final twist angle and rise value. The 3D initial model was a cylinder built by the RELION helix toolbox; 3D classification was performed several times to generate a proper reference map for 3D refinement. Three-dimensional refinement of the selected 3D classes with an appropriate reference was performed to obtain the final reconstruction. The final map of the G93A fibril was convergent with a rise of 4.88 Å and a twist angle of $-0.73°$. Postprocessing was performed to sharpen the map with a B factor of $-78.81$ Å$^2$. On the basis of the gold standard Fourier shell correlation (FSC) = 0.143 criteria, the overall resolution was reported as 3.09 Å. The statistics of cryo-EM data collection and refinement are shown in Table 1. For the D101N fibril assembled from bacterial-purified D101N, 25,344 fibrils were picked manually from 9310 micrographs, and 1024- and 686-pixel boxes were used to extract particles by a 90% overlap scheme. Two-dimensional classification of 1024-box size particles was used to calculate the initial twist angle. With regard to the helical rise, 4.8 Å was used as the initial value. The particles were extracted into 400-box sizes for further processing. After several iterations of 2D and 3D classifications, particles with the same morphology were picked out. Local searches of symmetry in 3D classification were used to determine the final twist angle and rise value. The 3D initial model was a cylinder built by the RELION helix toolbox; 3D classification was performed several times to generate a proper reference map for 3D refinement. Three-dimensional refinement of the selected 3D classes with an appropriate reference was performed to obtain the final reconstruction. The final map of the D101N fibril was convergent with a rise

of 4.82 Å and a twist angle of $-2.63°$. Postprocessing was performed to sharpen the map with a B factor of $-30.00$ Å$^2$. On the basis of the gold standard Fourier shell correlation (FSC) = 0.143 criteria, the overall resolution was reported as 2.92 Å. The statistics of cryo-EM data collection and refinement are shown in Table 1. For the G93A fibril assembled from N-terminally acetylated G93A, 13,314 fibrils were picked manually from 7799 micrographs, for the D101N fibril assembled from N-terminally acetylated D101N, 26,463 fibrils were picked manually from 5442 micrographs, for the wild-type fibril assembled from N-terminally acetylated wild-type SOD1, 4703 fibrils were picked manually from 2463 micrographs, and 1024-, 686-pixel, and 400-pixel boxes were used to extract particles by a 90% overlap scheme for 2D classification. After several iterations of 2D classification, particles with the same morphology were picked out.

## Atomic model building and refinement

Coot 0.8.9.2 (Emsley et al, 2010) was used to build de novo structures and modify the atomic models of the G93A fibril and the D101N fibril assembled from bacterial-purified G93A and D101N. Models with three adjacent layers were generated for structure refinement. The models were refined using the real-space refinement program in PHENIX 1.15.2 (Adams et al, 2010). All density map-related figures were prepared in Chimera 1.15. A ribbon representation of the structure of the SOD1 fibril was prepared in PyMol 2.3.

## Cell culture and transfection

SH-SY5Y neuroblastoma cells (catalog number GDC0210), HEK-293T cells (catalog number GDC0187), and HT-22 neuron cells (catalog number GDC0673) were obtained from the China Center for Type Culture Collection (CCTCC, Wuhan, China) and cultured in minimum essential media and Dulbecco's modified Eagle's medium (Gibco, Invitrogen), respectively, supplemented with 10% (v/v) fetal bovine serum (Gibco), 100 U/ml streptomycin, and 100 U/ml penicillin in 5% CO$_2$ at 37 °C. Expi293F suspension cells (catalog number A14635) were obtained from the Thermo Fisher Scientific (Gibco) and cultured in OPM-293 CD05 Medium (OPM), supplemented with 10% (v/v) OPM-293 ProFeed (OPM) in 5% CO$_2$ at 37 °C.

## Cell viability assays

SH-SY5Y cells, HEK-293T cells, or HT-22 neuron cells were plated in 96-well plates in minimum essential medium. After incubation for 24 h, wild-type SOD1 fibril seeds, G93A fibril seeds or D101N fibril seeds at a final concentration of 10 μM were added to the medium for 36 h. The MTT stock solution (5 mg/ml) was diluted with 1 × PBS (pH 7.4) and added to the wells for 4 h until formazan had formed in the cells. The final concentration of MTT was 0.5 mg/ml. Finally, the dark blue formazan crystals were dissolved in dimethyl sulfoxide, and the absorbance at 492 nm was measured using a Thermo Multiskan MK3 microplate reader (Thermo Fisher Scientific). The cells were incubated in medium containing 10% Cell Counting Kit-8 (CCK8) for 2–4 h, and the absorbance of the orange formazan was measured with a microplate reader at 450 nm. Cell viability was expressed as the percentage ratio of the

absorbance of wells containing the treated samples to that of wells containing cells treated with 20 mM Tris-HCl buffer (pH 7.4) containing 5 mM TCEP. The cell viability data, analyzed by using Origin Pro software version 8.0724 (Origin Laboratory), are expressed as the mean ± SD (with error bars) of values obtained from four independent experiments. All experiments were further confirmed by biological replicates.

## Statistical analysis

The data shown for each experiment were based on at least three technical replicates, as indicated in the individual figure legends. The data are presented as the mean ± SD, and $P$ values were determined using two-sided Student's $t$ tests. Differences were considered statistically significant when $P < 0.05$. All experiments were further confirmed by biological replicates.

## Data availability

The cryo-EM density maps and corresponding atomic coordinates for the human G93A SOD1 fibril have been deposited in the Electron Microscopy Data Bank (EMDB) under accession code EMD-60996 and in the Protein Data Bank (PDB) under accession code 9IYD. The cryo-EM density maps and corresponding atomic coordinates for the human D101N SOD1 fibril have been deposited in the EMDB under accession code EMD-60998 and in the PDB under accession code 9IYJ (https://www.rcsb.org/structure/9IYJ). Mass spectrometry data files (raw and search results) for Expi293F cell-purified wild-type SOD1 and its G93A and D101N variants have been deposited to the ProteomeXchange Consortium (https://proteomecentral.proteomexchange.org/cgi/GetDataset?ID=PXD066116) via the PRIDE partner repository with dataset identifier: PXD066116.

The source data of this paper are collected in the following database record: biostudies:S-SCDT-10_1038-S44319-025-00557-8.

## Peer review information

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

## Acknowledgements

Cryo-EM data were collected at the Core Facility of Wuhan University, China. All Cryo-EM data were processed at High Performance Computing Center in Interdisciplinary Research Center on Biology and Chemistry, Chinese Academy of Sciences, and in Wuhan University. We thank D Li (Cryo-EM Unit, Core Facility of Wuhan University) and X-N Li (Cryo-EM Unit, Core Facility of Wuhan University) for excellent assistance with cryo-EM; T O'Halloran (Northwestern University) for the gift of the human SOD1 plasmid; H Deng (Proteomics Facility at Technology Center for Protein Sciences, Tsinghua University) and M Han (Proteomics Facility at Technology Center for Protein Sciences, Tsinghua University) for protein MS analysis; L-T Hu (College of Life Sciences, Wuhan University) for technical assistance with TEM; and Y Wang (Institute of Biophysics, Chinese Academy of Sciences) for suggestions. This work was supported by funding from the Noncommunicable Chronic Diseases-National Science and Technology Major Project (Grant 2023ZD0507202) to L-Q Wang; the National Natural Science Foundation of China (NSFC) (Grants 32271326 and 32201040) to Y Liang and L-Q Wang; the National Key Research and Development Program of China (Grant 2024YFA1307301) and NSFC (Grant 32071212) to Y Liang; NSFC (Grants 82188101 and 22425704) to C Liu, Dr. Cong Liu is SANS Exploration Scholar; NSFC (Grant 12272276) to Z Wang; China Postdoctoral Science Foundation (Grants 2021TQ0252 and 2021M700103) to L-Q Wang; and the Key Project of Basic Research, Science and Technology Planning Project of Shenzen Municipality (Grant JCYJ20200109144418639) to L Zou.

## Author contributions

**Mu-Ya Zhang**: Formal analysis; Validation; Investigation; Visualization; Methodology; Writing—original draft. **Yeyang Ma**: Formal analysis; Investigation; Methodology. **Li-Qiang Wang**: Conceptualization; Formal analysis; Validation; Investigation; Visualization; Methodology; Writing—original draft. **Wencheng Xia**: Visualization; Methodology; Writing—original draft. **Xiang-Ning Li**: Data curation; Methodology. **Kun Zhao**: Methodology. **Jie Chen**: Resources; Project administration. **Dan Li**: Supervision. **Liangyu Zou**: Supervision; Funding acquisition. **Zhengzhi Wang**: Supervision; Funding acquisition; Methodology. **Cong Liu**: Conceptualization; Data curation; Formal analysis; Supervision; Funding acquisition; Investigation; Writing—review and editing. **Yi Liang**: Conceptualization; Data curation; Formal analysis; Supervision; Funding acquisition; Validation; Investigation; Visualization; Methodology; Writing—original draft; Project administration; Writing—review and editing.

Source data underlying figure panels in this paper may have individual authorship assigned. Where available, figure panel/source data authorship is listed in the following database record: biostudies:S-SCDT-10_1038-S44319-025-00557-8.

## Disclosure and competing interests statement

The authors declare no competing interests.

# Expanded View Figures

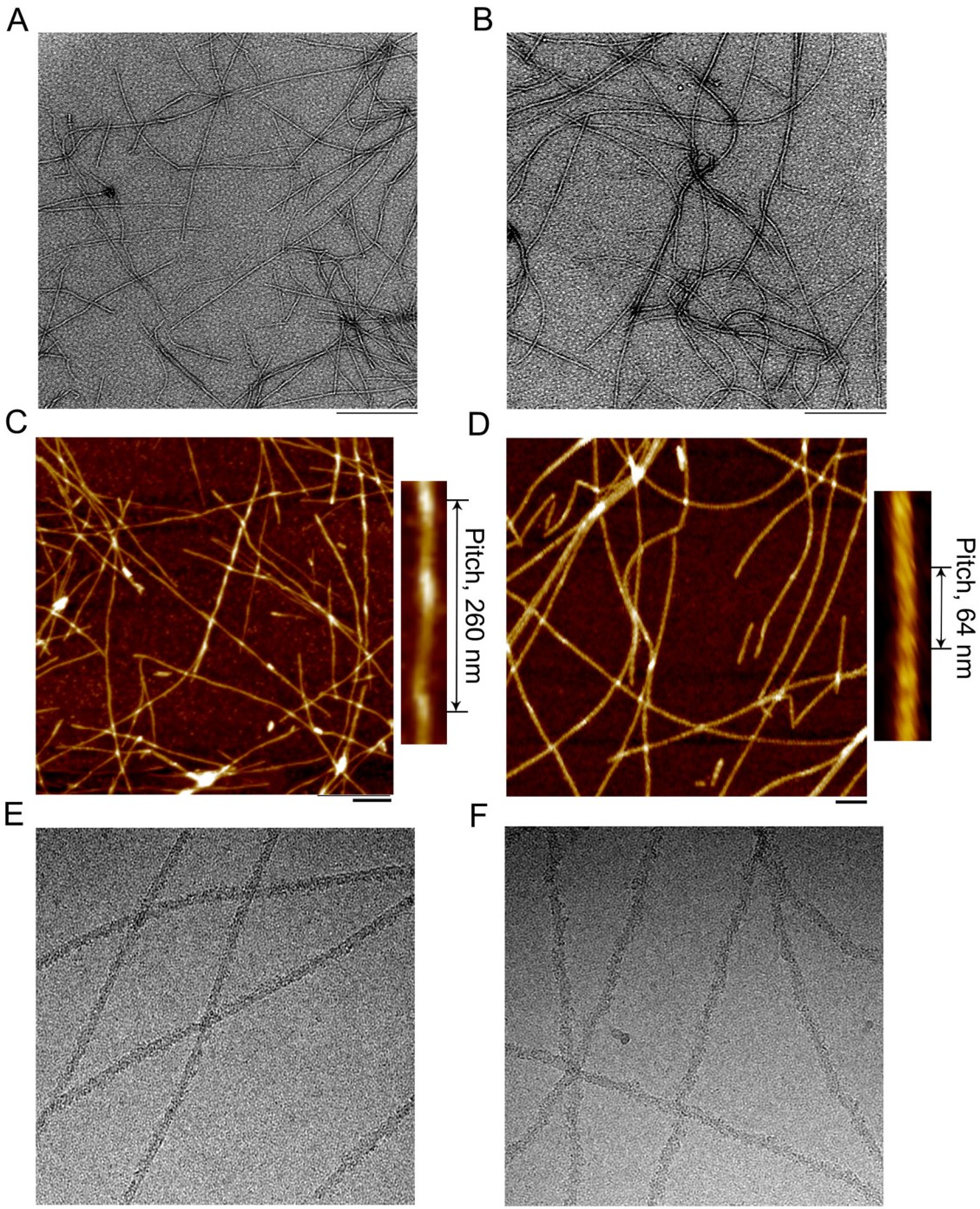

**Figure EV1. Comparison of the images of G93A fibrils and D101N fibrils.**

(A, B) Negative-staining TEM images of amyloid fibrils produced from the ALS-causing SOD1 mutant proteins G93A (A) and D101N (B). (C, D) AFM images of amyloid fibrils assembled from G93A (C) and D101N (D). The enlarged sections of (C) and (D) (right) showing the G93A fibril (C) and the D101N fibril (D) intertwined into a left-handed helix, with a helical pitch of 243 ± 11 nm and 64.3 ± 4.5 nm, respectively. The helical pitch was measured and expressed as the mean ± SD of values obtained in $n = 8$ biologically independent measurements. (E, F) Raw cryo-EM images of amyloid fibrils assembled from G93A (E) and D101N (F). The scale bars represent 200 nm (A–D) and 100 nm (E, F), respectively. Source data are available online for this figure.

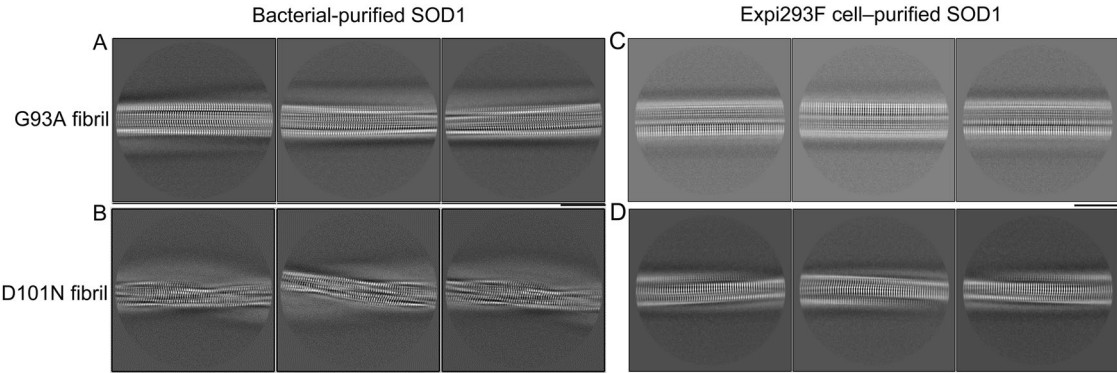

**Figure EV2. Comparison of the cryo-EM images of G93A fibril and D101N fibril.**

(A, B) Reference-free 2D class averages of the G93A fibril (A) and the D101N fibril (B) formed by bacterial-purified SOD1 mutants both showing a single protofilament intertwined. (C, D) Reference-free 2D class averages of the G93A fibril (C) and the D101N fibril (D) formed by Expi293F cell-purified SOD1 mutants showing a single protofilament less intertwined (C) and intertwined (D), respectively. Scale bars, 10 nm.

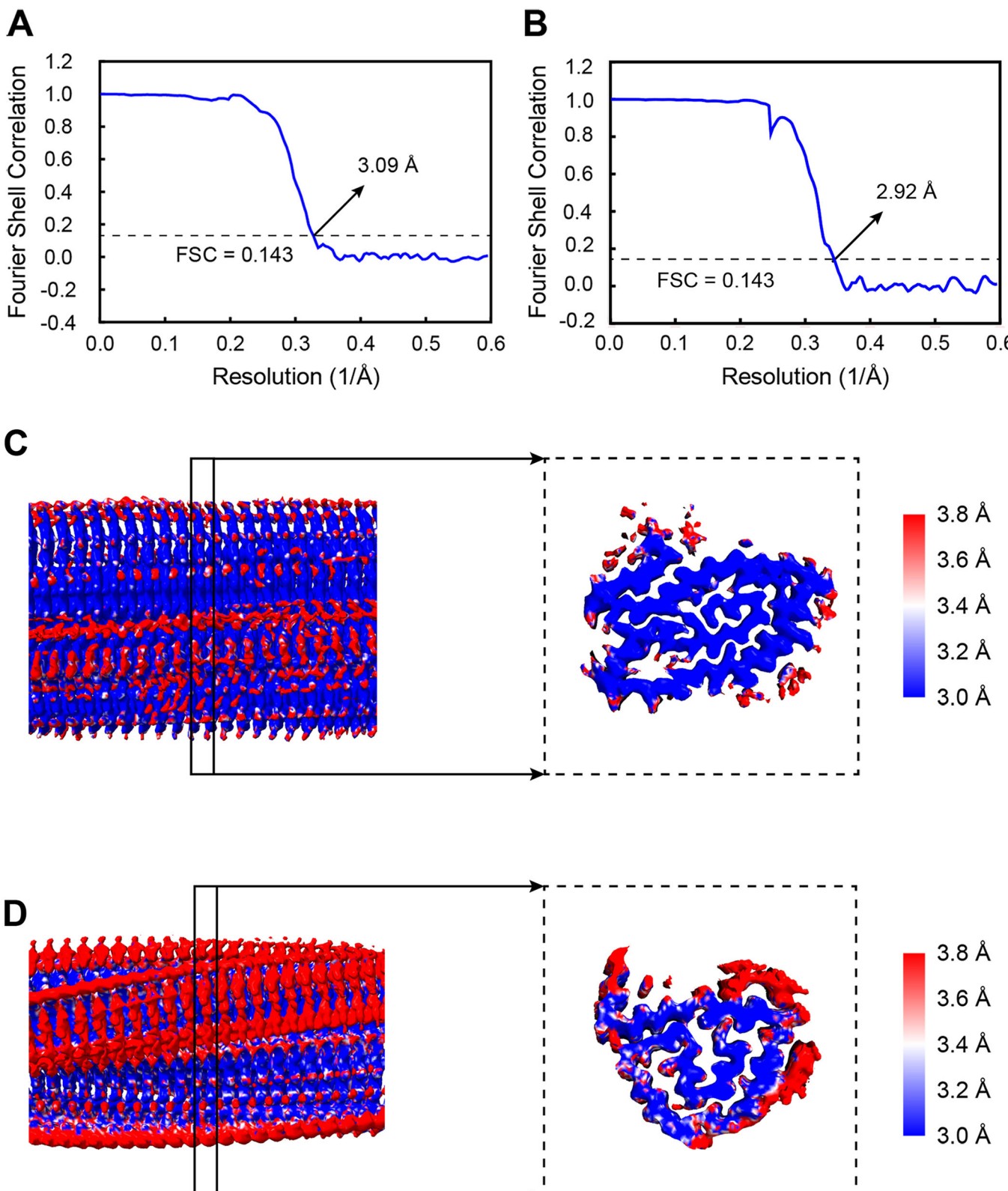

◀ **Figure EV3. Global (A and B) and local resolution (C and D) estimates for the reconstructions of G93A fibril and D101N fibril.**

(**A, B**) The reconstruction was reworked and gold-standard refinement was used for estimation of the density map resolution. The global resolutions of 3.09 Å for the G93A fibril (**A**) and 2.92 Å for the D101N fibril (**B**) were calculated using two Fourier shell correlation (FSC) curves (blue) cut-off at 0.143. (**C, D**) The density maps of G93A fibrils (**C**) and D101N fibrils (**D**) are colored according to local resolution estimated by ResMap. The enlarged cross sections show the left top view of the density maps of a single protofilament in the G93A fibril (**C**) and a single protofilament in the D101N fibril (**D**). The color keys on the right show the local structural resolution in angstroms (Å) and the colored maps indicate the local resolution ranging from 3.0 to 3.8 Å.

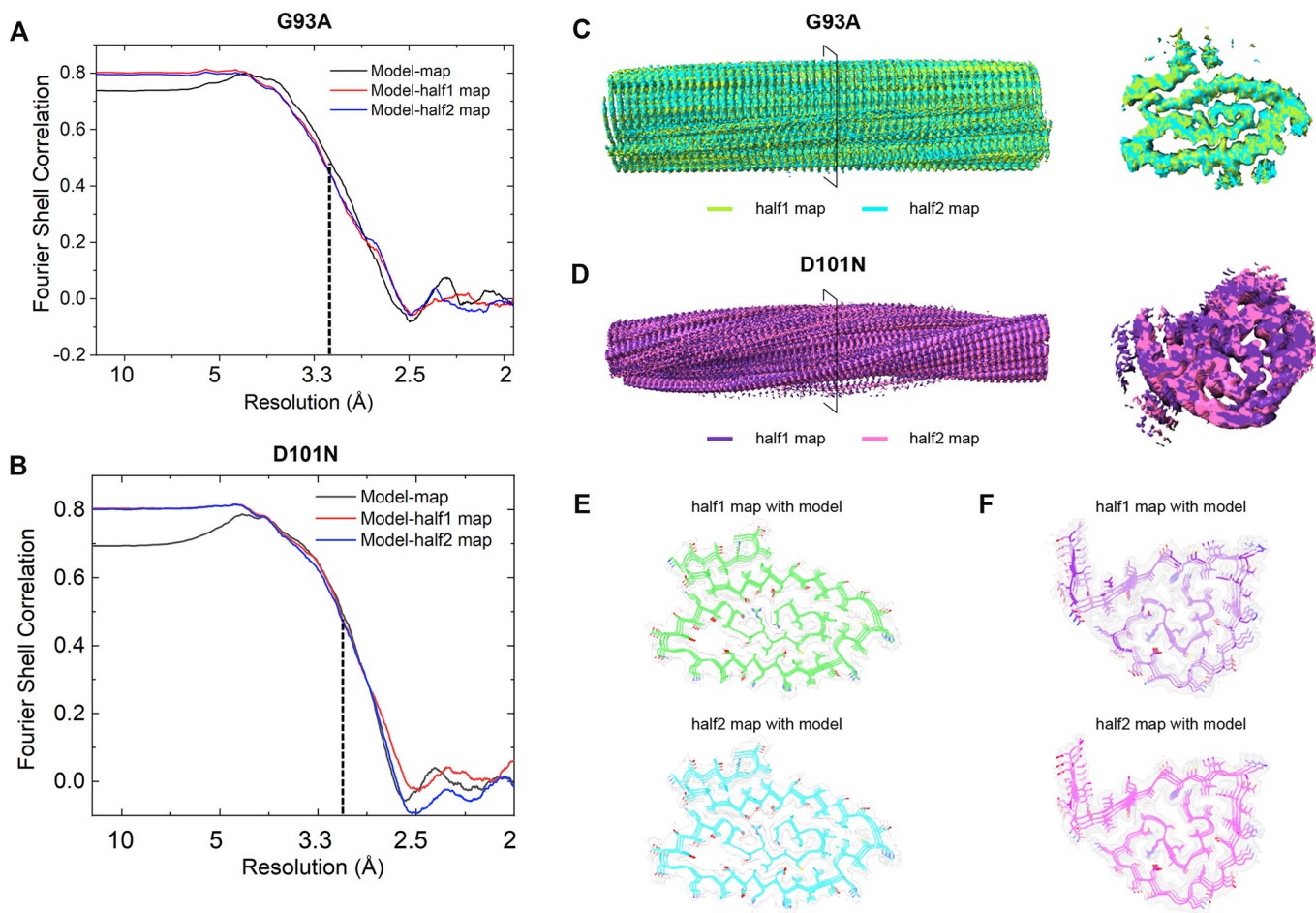

**Figure EV4.** For each structure of G93A fibril and D101N fibril, separate model refinements were performed against a single half-map, and the resulting model was compared with the other half-map to confirm the absence of overfitting.

(A, B) Fourier shell correlation (FSC) curves between the density map and the model. The FSC curves between the final refined model and the map reconstructed from all fibrils (black curve); between a model refined against the first half of the two independent half maps used for gold-standard FSC *versus* the reconstruction from that same half (red curve); and between a model refined against the first half of the two independent half maps *versus* the second independent half map (blue curve). The vertical lines at 3.09 Å (A) and 2.92 Å (B) indicate the highest resolution used in model refinement of the G93A fibril and the D101N fibril, respectively. These data convincingly demonstrate the absence of overfitting. (C, D) Comparison of the two optimized half-maps, half1 map (green and purple, respectively) and half2 map (cyan and magenta, respectively), from the G93A fibril (C) and the D101N fibril (D) 3D auto-refine process without mask, and the two half-maps match well. (E) Comparison of half1 map (green) and half2 map (cyan) of the G93A fibril with the atomic model overlaid. (F) Comparison of half1 map (purple) and half2 map (magenta) of the D101N fibril with the atomic model overlaid.

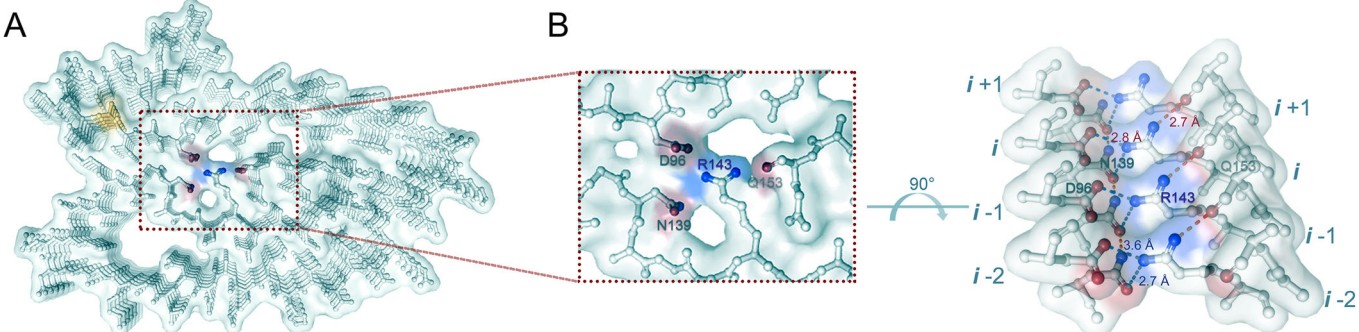

**Figure EV5.  Close-up view of the stick representation of the structure of G93A fibril stabilized by an intramolecular salt bridge and two hydrogen bonds.**

(**A**) A space-filled model overlaid onto a stick representation of the G93A fibril in which a single protofilament containing five molecular layers is shown in light green. The ALS-causing mutation site Ala93 is highlighted in yellow. Asp96/Arg143 pairs that form a new salt bridge are highlighted in red (oxygen atom in Asp96) and blue (nitrogen atom in Arg143), and the salt bridge region is magnified in (**B**). Asn139/Arg143 pairs and Gln153/Arg143 pairs that form hydrogen bonds are highlighted in red (oxygen atoms in Asn139 and Gln153) and blue (two nitrogen atoms in Arg143), and two hydrogen bond regions are also magnified in (**B**). (**B**) A magnified top view of the salt bridge region of a G93A protofilament, where Asp96/Arg143 pairs form a salt bridge. A side view (right) highlighting a strong salt bridge between Arg143 and Asp96 from the same molecular layer, with a distance of 2.7 Å (blue). Magnified top views of the two hydrogen bond regions of a G93A protofilament, where two pairs of amino acids (Asn139 and Arg143, and Gln153 and Arg143) form two hydrogen bonds. Two side views (right) highlighting a hydrogen bond between Arg143 and Asn139 from the same molecular layer, with a distance of 3.6 Å (blue), or between Arg143 from the molecular layer (*i*) and Gln153 from the adjacent molecular layer (*i* - 1), with a distance of 2.7 Å (red). The orientation of the structure of G93A fibril in (**A, B**) has been flipped 180° along X-axis so that it is the same as that in Fig. 2.

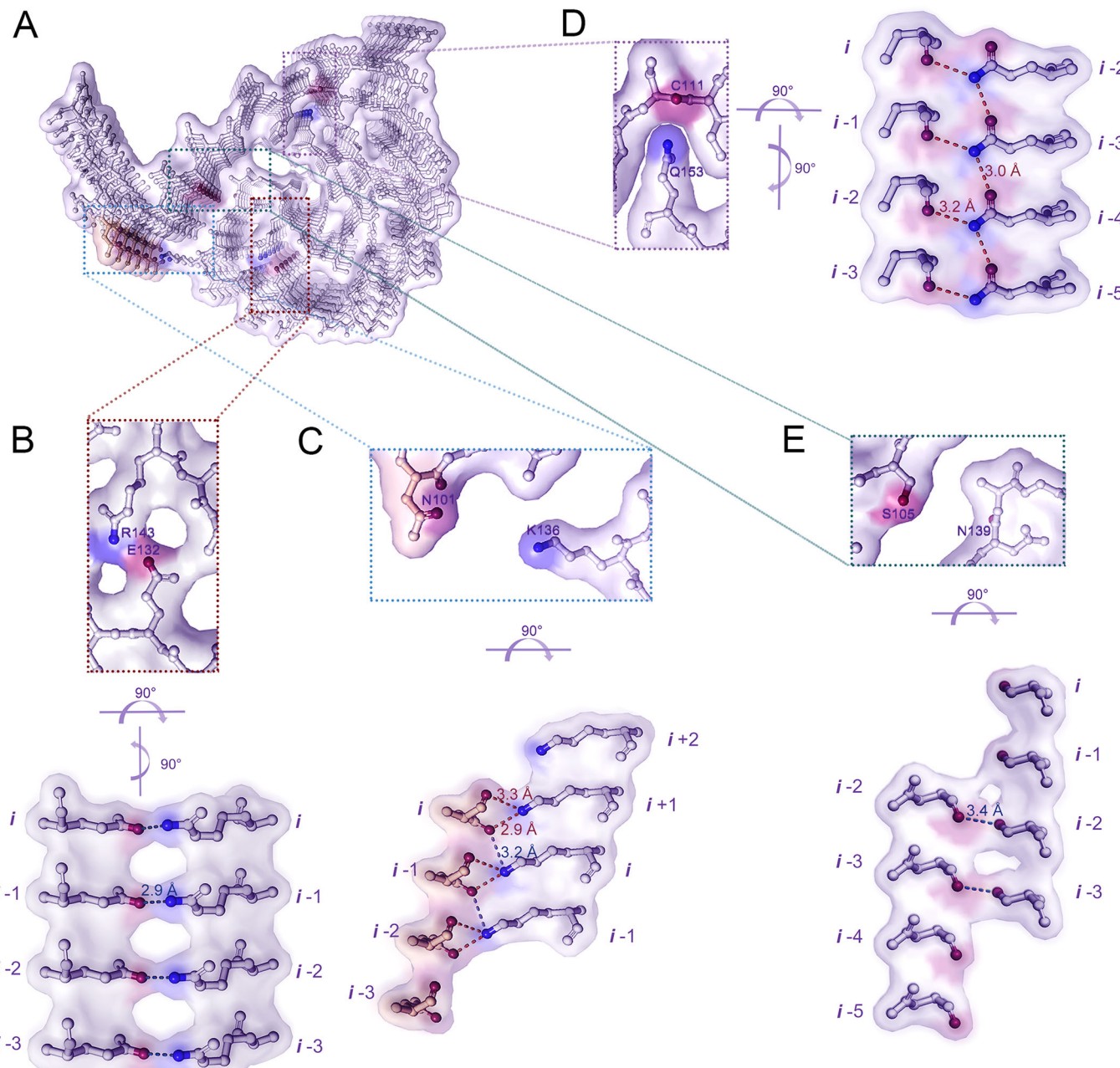

**Figure EV6. Close-up view of the stick representation of the structure of D101N fibril stabilized by an intramolecular salt bridge and four hydrogen bonds.**

(A) A space-filled model overlaid onto a stick representation of the D101N fibril in which a single protofilament is shown in light purple. The ALS-causing mutation site Asn101 is highlighted in gold. Asp132/Arg143 pairs that form a new salt bridge are highlighted in red (oxygen atom in Asp132) and blue (nitrogen atom in Arg143), and the salt bridge region is magnified in (B). Asn101/Lys136 pairs and Cys111/Gln153 pairs that form hydrogen bonds are highlighted in red (two oxygen atoms in Asn101 and oxygen atoms in Cys111) and blue (nitrogen atoms in Lys136 and Gln153), Asn139/Ser105 pairs that form a hydrogen bond are highlighted in red (oxygen atom in Asn139) and red (hydroxy group in Ser105), and three hydrogen bond regions are magnified in (C—E). (B) A magnified top view of the salt bridge region of a D101N protofilament, where Asp132/Arg143 pairs form a salt bridge. A side view (bottom) highlighting a strong salt bridge between Arg143 and Asp132 from the same molecular layer, with a distance of 2.9 Å (blue). (C) A magnified top view of a hydrogen bond region of a D101N protofilament, where Asn101/Lys136 pairs form two hydrogen bonds. A side view (bottom) highlighting a hydrogen bond between the main chain of Asn101 from the molecular layer ($i$) and Lys136 from the adjacent molecular layer ($i + 1$), with a distance of 2.9 Å (red), or between Asn101 from the molecular layer ($i$) and Lys136 from the adjacent molecular layer ($i + 1$), with a distance of 3.3 Å (red). (D) A magnified top view of a hydrogen bond region of a D101N protofilament, where Cys111/Gln153 pairs form a hydrogen bond. A side view (right) highlighting a hydrogen bond between the main chain of Cys111 from the molecular layer ($i$) and Gln153 from the molecular layer ($i - 2$), with a distance of 3.2 Å (red). (E) A magnified top view of a hydrogen bond region of a D101N protofilament, where Asn139/Ser105 pairs form a hydrogen bond. A side view (bottom) highlighting a hydrogen bond between the main chain of Asn139 and the hydroxy group in Ser105 from the same molecular layer, with a distance of 3.4 Å (blue).

