## [Peer Review File · EMBO Reports]

Distinct amyloid fibril structures formed by ALS-causing SOD1 mutants G93A and D101N

Mu-Ya Zhang, Yeyang Ma, Li-Qiang Wang, Wencheng Xia, Xiang-Ning Li, Kun Zhao, Jie Chen, Dan Li, Liangyu Zou, Zhengzhi Wang, Cong Liu, and Yi Liang

Corresponding authors: Yi Liang (liangyi@whu.edu.cn), Cong Liu (liulab@sioc.ac.cn), Li-Qiang Wang (wangliqiang@whu.edu.cn)

Review Timeline:

Submission Date:	24th Oct 24
Editorial Decision:	5th Dec 24
Revision Received:	11th May 25
Editorial Decision:	27th Jun 25
Revision Received:	22nd Jul 25
Accepted:	7th Aug 25

Transaction Report:

Dear Prof. Liang

Thank you for the submission of your research manuscript to our journal. We have now received the full set of referee reports that is copied below.

As you will see, the referees acknowledge that the findings are potentially interesting, but they also raise a number of important concerns regarding the structural data and the toxicity data. While the cryo-EM structures as such seem convincing, they were obtained using SOD1 proteins purified from *E. coli*. These lack the N-terminal acetylation (ref #2) and the structures appear to be different from earlier reported structures (ref #3). Moreover, toxicity assays were not performed on neuron cells and the relevance to ALS remains unclear.

I have discussed these concerns, the strengths and limitations of the study with the referees and the editorial team and based on these consultations, I would like to invite you to revise your manuscript for EMBO Reports. Please provide further indicators/parameters of toxicity on neuroblastoma cells and/or perform toxicity assays in neurons, if this is possible. Please also discuss the limitations using purified proteins as well as the divergence from the published structure and the "three key region model".

Please let me know in case you disagree, and we can discuss the exact revision requirements further, also in a video chat, if you like.

Please address all referee concerns in a complete point-by-point response. Acceptance of the manuscript will depend on a positive outcome of a second round of review. It is EMBO Reports policy to allow a single round of revision only and acceptance or rejection of the manuscript will therefore depend on the completeness of your responses included in the next, final version of the manuscript.

We realize that it is difficult to revise to a specific deadline. In the interest of protecting the conceptual advance provided by the work, we recommend a revision within 3 months (March 5th, 2025). Please discuss the revision progress ahead of this time with the editor if you require more time to complete the revisions.

I am also happy to discuss the revision further via e-mail or a video call, if you wish.

*******IMPORTANT NOTE:**

We perform an initial quality control of all revised manuscripts before re-review. Your manuscript will FAIL this control and the handling will be delayed IN CASE the following APPLIES:

- 1) A data availability section providing access to data deposited in public databases is missing. If you have not deposited any data, please add a sentence to the data availability section that explains that.
- 2) Your manuscript contains statistics and error bars based on $n=2$. Please use scatter blots in these cases. No statistics should be calculated if $n=2$.

When submitting your revised manuscript, please carefully review the instructions that follow below. Failure to include requested items will delay the evaluation of your revision. *****

2) individual production quality figure files as .eps, .tif, .jpg (one file per figure).

Please download our Figure Preparation Guidelines (figure preparation pdf) from our Author Guidelines pages <https://www.embopress.org/page/journal/14693178/authorguide> for more info on how to prepare your figures.

3) a .docx formatted letter INCLUDING the reviewers' reports and your detailed point-by-point responses to their comments. As

part of the EMBO Press transparent editorial process, the point-by-point response is part of the Review Process File (RPF), which will be published alongside your paper.

4) a complete author checklist, which you can download from our author guidelines (<<https://www.embopress.org/page/journal/14693178/authorguide>>). Please insert information in the checklist that is also reflected in the manuscript. The completed author checklist will also be part of the RPF.

5) Please note that all corresponding authors are required to supply an ORCID ID for their name upon submission of a revised manuscript (<<https://orcid.org/>>). Please find instructions on how to link your ORCID ID to your account in our manuscript tracking system in our Author guidelines (<<https://www.embopress.org/page/journal/14693178/authorguide#authorshipguidelines>>)

6) We replaced Supplementary Information with Expanded View (EV) Figures and Tables that are collapsible/expandable online. A maximum of 5 EV Figures can be typeset. EV Figures should be cited as 'Figure EV1, Figure EV2' etc... in the text and their respective legends should be included in the main text after the legends of regular figures.

7) Data Availability section: please include URLs that directly resolve to the datasets at EMD/PDB.

Additional information on source data and instruction on how to label the files are available <<https://www.embopress.org/page/journal/14693178/authorguide#sourcedata>>.

10) Figure legends and data quantification:
The following points must be specified in each figure legend:

- the name of the statistical test used to generate error bars and P values,
- the number (n) of independent experiments (please specify technical or biological replicates) underlying each data point,
- the nature of the bars and error bars (s.d., s.e.m.)
- If the data are obtained from n {less than or equal to} 5, show the individual data points in addition to the SD or SEM.
- If the data are obtained from n {less than or equal to} 2, use scatter blots showing the individual data points.

See also the guidelines for figure legend preparation:
<https://www.embopress.org/page/journal/14693178/authorguide#figureformat>

11) Our journal encourages inclusion of *data citations in the reference list* to directly cite datasets that were re-used and obtained from public databases. Data citations in the article text are distinct from normal bibliographical citations and should directly link to the database records from which the data can be accessed. In the main text, data citations are formatted as follows: "Data ref: Smith et al, 2001" or "Data ref: NCBI Sequence Read Archive PRJNA342805, 2017". In the Reference list, data citations must be labeled with "[DATASET]". A data reference must provide the database name, accession number/identifiers and a resolvable link to the landing page from which the data can be accessed at the end of the reference. Further instructions are available at <<https://www.embopress.org/page/journal/14693178/authorguide#referencesformat>>.

12) All Materials and Methods need to be described in the main text using our 'Structured Methods' format. According to this format, the Methods section includes a Reagents and Tools Table (listing key reagents, experimental models, software and relevant equipment and including their sources and relevant identifiers) followed by a Methods and Protocols section describing the methods, ideally using a step-by-step protocol format. The aim is to facilitate adoption of the methodologies across labs. Please download and fill our Reagents and Tools Table template (.docx), which you can find in our author guidelines: <https://www.embopress.org/page/journal/14693178/authorguide#structuredmethods>.

An example of a Method paper with Structured Methods can be found here: <https://www.embopress.org/doi/10.15252/msb.20178071>.

13) As part of the EMBO publication's Transparent Editorial Process, EMBO Reports publishes online a Review Process File to accompany accepted manuscripts. This File will be published in conjunction with your paper and will include the referee reports, your point-by-point response and all pertinent correspondence relating to the manuscript.

Yours sincerely,

Referee #1:

Zhang and colleagues report high-resolution structures of two mutant types of SOD1 amyloid fibrils generated in vitro. These mutants are linked to genetic forms of ALS, and as shown here, can form fibril conformers that differ from each other and from those formed from wild-type SOD1. The authors also provide evidence that one of these fibril types (G93A) was more toxic than the type formed by the other mutant (D101N). The data are convincing and well-presented (for the most part). The results should be of interest to the ALS field. I have only a few minor suggestions to improve the manuscript.

- 1) Middle of p. 9: theses should be these.
- 2) Top of p. 15: the cells were described as being incubated for 1.5 d in a Tris buffer containing fibrils, but presumably this buffer was diluted into tissue culture medium as well for this incubation (or the cells would have died)? Please check.
- 3) The presentation as a whole is a bit more repetitive than it needs to be, and would be more readily digestible by readers if redundancies between different sections were minimized.

Referee #2:

1. Decline publication.
2. This manuscript describes a single key finding, that there is distinct toxicity between wild-type and G93A ALS causing mutant of human SOD1 (G93A and D101N were studied). While such a finding would satisfy the journal's requirement, I feel the data quality does not achieve the necessary rigor. While the finding may be of general interest to the molecular biology community, I do not feel the data is robust enough to proceed.
3. Clarity is very good. Figure 4 clearly shows the suggested difference in toxicity but I do not feel the difference is significant for such a conclusion.
4. A significant flaw arises in this work. Human SOD1 is post-translationally modified, removing the initiating Met and acetylating the N-terminus at A2. In this study, the proteins are expressed in *E. coli* where the N-terminus is not acetylated. In some examples this point is probably not significant, however in the case of SOD1 and ALS this is not the case. Any covalent

difference in the structure can be critical to propensity to form fibrils. The existence of the A4V ALS-causing mutant drives home this point for SOD1 especially at the N-terminus of the protein. The SOD1 protein expressed in E. coli is not the same as the same SOD1 sequence expressed in humans (or yeast, as in the Valentine lab). I argue that mass spectrometry should be used to characterize the covalent state of the intact protein to highlight the difference (or not, if I am wrong). The problem now is that this work compares wild type to 2 ALS mutants in the wrong background, which might behave differently to the human-expressed form. Is this a fatal flaw? It is hard to ignore the incorrect background at this point. I think the work should be repeated in the correct background - a eukaryote that acetylates the N terminus after removal of Met 1 (such as yeast).

The activity measurements warrant some discussion. WT fibrils and G93A fibrils look different in the CCK8 assay in SH-SY5Y, less so in the 3 other assays. D101N was not significantly different from WT. The results of the cell viability assays are to me the weakest part of the data presented. Yet we are asked whether the paper is publishable based upon a conclusion from the activity measurements. I am inclined to decline publication based on this data. Since the experiments were performed in the wrong background I am again left to conclude that these measurements must be repeated with SOD1 with an acetylated A2 N-terminus.

5. I think these comments are clear.

Referee #3:

This manuscript reported two cryo-EM structures of amyloid fibrils formed by G93A and D101N mutants of SOD1 protein. These mutations give rise to amyloid fibrils with distinct structures compared to native SOD1 fibrils. By toxicity assays on the SH-SY5Y neuroblastoma cells and HEK-293T cells, they found G93A fibrils are significantly more toxic than those formed by D101N, which do not show a marked increase in toxicity compared to wild-type SOD1 fibrils. The biggest weakness of this manuscript is the structures are of in vitro aggregated SOD1 protein, and the structures are very different from "three key region model" proposed from the study on the transgenic mice, therefore the relevance to the human ALS diseases remains unknown. Toxicity assay would be more convincing if it was done on neuron cells and more indicators were studied except for viability. Nevertheless, the quality of Cryo-EM structures is good and novel. The mutants they studied are important and the manuscript should be of interest to the ALS field.

Here are the details for the revision:

1. The SDS-PAGE gels of SOD1 protein before and after in vitro aggregation should be added to show the intact of the protein, because it raised concern that the protein was degraded since only C-terminal protein was incorporated in the ordered core of structures and no proteinase inhibitors were used during apo-SOD1 protein purification.
2. The accuracy of using NanoDrop to measure the concentration of the fibrils is questionable. ELISA or MSD assays could be used for accurately measuring the concentration of monomers denatured from the fibrils. The length of the fibrils may affect the results as well. NS-EM images of the input fibrils for WT, G93A and D101N fibrils should be shown.
3. The orientation of the G93A structure in FigS4 is different from that in Fig2, which may cause confusion. It should be flipped 180° along X-axis.
4. For each structure, separate model refinements should be performed against a single half-map, and the resulting model should be compared with the other half-map to confirm the absence of overfitting.
5. The cross-sectional view of 3D map in Fig 1A and B could be improved by showing the cross-sectional view of one rung. The indicated pitch range indicated in Figure 1E is not accurate. The colour bar in Fig 3G is reversed. Positions of G93A and D101N should be labelled in Fig 5B,C,D

Referee #1:

Summary:

Zhang and colleagues report high-resolution structures of two mutant types of SOD1 amyloid fibrils generated in vitro. These mutants are linked to genetic forms of ALS, and as shown here, can form fibril conformers that differ from each other and from those formed from wild-type SOD1. The authors also provide evidence that one of these fibril types (G93A) was more toxic than the type formed by the other mutant (D101N). The data are convincing and well-presented (for the most part). The results should be of interest to the ALS field.

We sincerely thank the referee for recognizing the significance of our work. The referee's suggestion is very valuable for us to improve our manuscript. Right now, we have revised and added the following sentences into the revised manuscript, as followed the advice of referee #1. Given that bacterial-purified G93A and D101N do form fibril conformers that differ from each other (Figs. 2 and 3), ... (page 15, lines 1-2). In summary, on the one hand, two SOD1 metal-binding mutants, G93A and D101N, ..., form fibril conformers that differ from each other and from those formed from wild-type SOD1, as revealed by cryo-EM; (page 25, lines 6-12).

I have only a few minor suggestions to improve the manuscript.

Comment #1 • Middle of p. 9: theses should be these.

REPLY: Thank for pointing this out. Right now, we have replaced “theses” with “these” in our manuscript (page 9, line 4), as followed referee's suggestion.

Comment #2 • Top of p. 15: the cells were described as being incubated for 1.5 d in a Tris buffer containing fibrils, but presumably this buffer was diluted into tissue culture medium as well for this incubation (or the cells would have died)? Please check.

REPLY: We apologize for this confusion. We totally agree that the statement “SH-SY5Y neuroblastoma cells and HEK-293T cells were cultured for 1 day and then incubated with 20 mM Tris-HCl buffer (pH 7.4) containing 5 mM TCEP (control), 10 μM wild-type SOD1 fibril seeds, 10 μM G93A fibril seeds, or 10 μM D101N fibril seeds, respectively, for 1.5 days ...” in the previous version is confusing. To clarify

and elaborate upon, we have now reworded and added the following sentences into the revised manuscript. *SH-SY5Y neuroblastoma cells, HEK-293T cells, and HT-22 neuron cells were cultured for 1 day, then 20 mM Tris-HCl buffer (pH 7.4) containing 5 mM TCEP (control), wild-type SOD1 fibril seeds, G93A fibril seeds, or D101N fibril seeds was diluted into tissue culture medium, respectively, and the cells were cultured for 1.5 days and ... (page 15, lines 6-10). The cells were cultured for 1 day, then 20 mM Tris-HCl buffer (pH 7.4) containing 0 μM SOD1 fibril seeds, wild-type SOD1 fibril seeds, G93A fibril seeds, or D101N fibril seeds was diluted into tissue culture medium, respectively, and the cells were further cultured for 1.5 days. The final concentration of fibril seeds from bacterial-purified wild-type SOD, G93A, and D101N was slightly smaller than 10 μM (page 52, lines 7-12, Legend of Fig. 4).*

Comment #3 • The presentation as a whole is a bit more repetitive than it needs to be, and would be more readily digestible by readers if redundancies between different sections were minimized.

REPLY: We sincerely thank the referee for this important suggestion to improve our manuscript. We totally agree that the presentation of the previous manuscript as a whole is a bit more repetitive than it needs to be. According to the advice of referee #1, we have now made the text more concise and redundancies between different sections are minimized. We hope that it would be more readily digestible by readers. For instance, "... demonstrating distinct structural formations for each mutant" in the previous version have been replaced by "... *demonstrating unique structural features for each mutant*" in the revised manuscript (page 3, lines 8-9 up, Abstract). "We demonstrated that G93A and D101N form distinct amyloid fibril structures ..." in the previous version have been replaced by "*We demonstrated that G93A and D101N form different amyloid fibril structures ...*" in the revised manuscript (page 9, lines 8-9 up). "Given that two ALS-causing SOD1 mutants, G93A and D101N, do form distinct amyloid fibril structures (Figs. 2 and 3), ..." in the previous version have been reworded into "*Given that bacterial-purified G93A and D101N do form fibril conformers that differ from each other (Figs. 2 and 3), ...*" in the revised manuscript (page 15, lines 1-2). "..., form distinct amyloid fibril structures, as revealed by cryo-EM;" in

the previous version have been reworded into “..., form fibril conformers that differ from each other and from those formed from wild-type SOD1, as revealed by cryo-EM;” in the revised manuscript (page 25, lines 11-12).

Referee #2:

Summary:

Decline publication. This manuscript describes a single key finding, that there is distinct toxicity between wild-type and G93A ALS causing mutant of human SOD1 (G93A and D101N were studied). While such a finding would satisfy the journal's requirement, I feel the data quality does not achieve the necessary rigor. While the finding may be of general interest to the molecular biology community, I do not feel the data is robust enough to proceed. Clarity is very good. Figure 4 clearly shows the suggested difference in toxicity but I do not feel the difference is significant for such a conclusion.

We sincerely apologize that the previous version of our work did not meet the referee's high standards. Still, we appreciate the referee's insightful comments, which are very valuable to overcome the shortcomings of the previous version of our work. We have fully addressed the comments from referee #2 in our revised manuscript and have made every effort to improve the manuscript markedly. Right now, we have provided the toxicity assay data of amyloid fibrils from Expi293F cell-purified G93A, D101N, and wild-type SOD1 including in neuron cells (Fig. 6A–F, page 62) and documented the morphology of these amyloid fibrils formed by N-terminally acetylated G93A and D101N using TEM and cryo-EM (Fig. 5A–I, page 61) as well as cryo-EM 2D classes (Fig. EV2C,D, page 66, Expanded View Figures), and do hope that the new data quality could achieve the necessary rigor.

Comment #1 • A significant flaw arises in this work. Human SOD1 is post-translationally modified, removing the initiating Met and acetylating the N-terminus at A2. In this study, the proteins are expressed in *E. coli* where the N-terminus is not acetylated. In some examples this point is probably not significant, however in the case of SOD1 and ALS this is not the case. Any covalent difference in the structure can be critical to propensity to form fibrils. The existence of the A4V ALS-causing

mutant drives home this point for SOD1 especially at the N-terminus of the protein. The SOD1 protein expressed in *E. coli* is not the same as the same SOD1 sequence expressed in humans (or yeast, as in the Valentine lab). I argue that mass spectrometry should be used to characterize the covalent state of the intact protein to highlight the difference (or not, if I am wrong). The problem now is that this work compares wild type to 2 ALS mutants in the wrong background, which might behave differently to the human-expressed form. Is this a fatal flaw? It is hard to ignore the incorrect background at this point. I think the work should be repeated in the correct background - a eukaryote that acetylates the N terminus after removal of Met 1 (such as yeast).

The activity measurements warrant some discussion. WT fibrils and G93A fibrils look different in the CCK8 assay in SH-SY5Y, less so in the 3 other assays. D101N was not significantly different from WT. The results of the cell viability assays are to me the weakest part of the data presented. Yet we are asked whether the paper is publishable based upon a conclusion from the activity measurements. I am inclined to decline publication based on this data. Since the experiments were performed in the wrong background, I am again left to conclude that these measurements must be repeated with SOD1 with an acetylated A2 N-terminus.

REPLY: We sincerely thank referee #2 for this important suggestion that we should repeat our experiments in the correct background - a eukaryote that acetylates the N terminus after removal of Met 1 (such as yeast)!! We also sincerely thank the referee for his (her) expert suggestion that we should express recombinant full-length wild-type human SOD1 and its G93A and D101N variants in mammalian cells (or yeast, as in the Valentine lab)!! Indeed, the recombinant full-length wild-type human SOD1 and its G93A and D101N variants we used previously were expressed in *Escherichia coli* where the N-terminus is not acetylated. According to the advice of referee #2, we have now provided the toxicity assay data of amyloid fibrils from Expi293F cell-purified G93A, D101N, and wild-type SOD1 including in neuron cells (Fig. 6A–F, page 62) and documented the morphology of these amyloid fibrils formed by N-terminally acetylated G93A and D101N using TEM and cryo-EM (Fig. 5A–I, page 61) as well as cryo-EM 2D classes (Fig. EV2C,D, page 66, Expanded View Figures). We have added the following subsection titled “*Comparison of the images of G93A*

fibrils and D101N fibrils formed by N-terminally acetylated SOD1 mutants” into the Results section of the revision.

Human SOD1 is post-translationally modified, removing the initiating Met and acetylating the N-terminus at Ala1. We expand on the limitations of the aforementioned study. First, the recombinant SOD1 mutants we used previously are expressed in Escherichia coli where the N-terminus is not acetylated. Second, the aforementioned study does not consider the influence of N-terminal acetylation of SOD1 on the structures and functions of G93A fibrils and D101N fibrils though any covalent difference in the structure can be critical to propensity to form fibrils. Such an influence could be especially pronounced for the case of the A4V ALS-causing mutant, whose mutation site Val4 is at the N-terminus of the protein. Therefore, the impact of our studies would be considerably enhanced by testing our basic conclusions in other experimental systems that contain the usually N-terminally acetylated SOD1 (Figs. 5, 6, and EV2C,D). To investigate this, we expressed and purified recombinant full-length wild-type human SOD1 and its G93A and D101N variants in Expi293F cells where the initiating Met was removed and the N-terminus at Ala1 was acetylated, and identified N-terminal acetylation of the mammalian cell-purified G93A (Fig. 5A), D101N (Fig. 5B), and wild-type SOD1 (Fig. 5C) using mass spectrometry (MS). Analysis of the b-ions in Fig. 5A–C indicated +42.01037, +42.01014, and +42.01014 Da mass shifts representing the addition of an acetyl group to Ala1, demonstrating N-terminal acetylation in the mammalian cell-purified G93A, D101N, and wild-type SOD1.

We produced amyloid fibrils from recombinant, full-length apo human SOD1 (residues 1 to 153) with G93A mutation or D101N mutation overexpressed in Expi293F cells by incubating the purified apoproteins in 20 mM Tris-HCl buffer (pH 7.4) containing 5 mM TCEP and shaking at 37 °C for 34 to 36 hours (see methods). Amyloid fibrils formed by G93A and D101N under these reducing conditions were also concentrated to ~30 μM in a centrifugal filter (Millipore) and examined by electron microscopy without further treatment. Negative-staining TEM images showed that the apo forms of N-terminally acetylated G93A, D101N, and wild-type SOD1 all formed homogeneous and unbranched fibrils under reducing conditions (Fig. 5D–F). The cryo-EM micrographs (Fig. 5G–I) and 2D class average images obtained using RELION3.1 (Scheres, 2020) showed that the

G93A fibril and the *D101N* fibril formed by N-terminally acetylated *SOD1* mutants were composed of a single protofilament less intertwined (Fig. EV2C) and intertwined (Fig. EV2D), respectively. The 2D class average images showed that at least *in vitro*, N-terminally acetylated *G93A* and *D101N* form fibril conformers that differ from each other and from those formed from bacterial-purified *G93A* and *D101N* (Fig. EV2A–D).

Together, these results demonstrate that N-terminally acetylated *G93A* and *D101N* mutants also form distinct amyloid fibril structures, confirming our basic conclusion in other experimental systems that contain the usually N-terminally acetylated *SOD1*.

(The above paragraphs, pages 16-18).

Figure 5. Comparison of the images of *G93A* fibrils and *D101N* fibrils formed by N-terminally acetylated *SOD1* mutants.

Recombinant full-length wild-type human *SOD1* and its *G93A* and *D101N* variants were expressed and purified in Expi293F cells where the initiating Met was removed and the N-terminus at Ala1 was acetylated. (A–C) Identification of N-terminal

acetylation of the mammalian cell-purified G93A (A), D101N (B), and wild-type SOD1 (C) using mass spectrometry (MS). The Coomassie Blue-stained gels of SDS-PAGE of N-terminally acetylated wild-type SOD1 and its G93A and D101N variants were scissored out, chopped, trypsinized, and then analyzed with nano-LC-MS/MS. An MS² analysis of parent peptide A¹TKAVCVL⁸ digested by trypsin. (A) Analysis of the b-ion (b₁⁺¹) indicates +42.01037 Da mass shift (114.05528 minus 1.0078 = 113.04748 Da, 113.04748 minus 71.03711 = 42.01037 Da) representing the addition of an acetyl group to Ala1, demonstrating N-terminal acetylation in the mammalian cell-purified G93A. (B and C) Analysis of the b-ion (b₁⁺¹) indicates +42.01014 Da mass shift (114.05505 minus 1.0078 = 113.04725 Da, 113.04725 minus 71.03711 = 42.01014 Da) representing the addition of an acetyl group to Ala1, demonstrating N-terminal acetylation in the mammalian cell-purified D101N (B) and wild-type SOD1 (C). (D–F) Negative-staining TEM images of amyloid fibrils produced from N-terminally acetylated G93A (D), D101N (E), and wild-type SOD1 (F). (G–I) Raw cryo-EM images of amyloid fibrils assembled from N-terminally acetylated G93A (G), D101N (H), and wild-type SOD1 (I). The scale bars represent 200 nm (D–F) and 100 nm (G–I), respectively.

Figure EV2. Comparison of the cryo-EM images of G93A fibril and D101N fibril.

(A and B) Reference-free 2D class averages of the G93A fibril (A) and the D101N fibril (B) formed by bacterial-purified SOD1 mutants both showing a single protofilament intertwined. (C and D) Reference-free 2D class averages of the G93A fibril (C) and the D101N fibril (D) formed by Expi293F cell-purified SOD1 mutants showing a single protofilament less intertwined (C) and intertwined (D), respectively. Scale bars, 10 nm.

We have added the following sentences into the Methods section of the revision, including a subsection titled “LC-MS/MS analysis” (pages 29-30). *The target genes for wild-type SOD1 and its mutants G93A and D101N were subcloned into the pcDNA3.4, and the SOD1 plasmids were transfected into Expi293F cells using PEI max transfection reagent (Polysciences 24765-100). After 4 days of transfection, the cells were harvest by centrifugation at 8,000 rpm at 4 °C for 6 min and then washed with 1 × PBS (pH 7.4) for three times. The cells were resuspended with 1 × PBS (pH 7.4) and sonicated at 30 W for 15 min. The supernatant was collected after centrifugation at 10,000 × g for 15 min, then wild-type SOD1 and its G93A and D101N variants were captured by Ni Smart Beads and eluted in imidazole (page 28, lines 1-9). The apo forms of Expi293F cell-purified full-length wild-type human SOD1 and its G93A and D101N variants were incubated in 20 mM Tris-HCl buffer (pH 7.4) containing 5 mM TCEP and shaken at 37 °C for 34 to 36 hours, after which the SOD1 fibrils were collected (page 30, lines 6-9 up). G93A fibrils and D101N fibrils assembled from bacterial-purified G93A and D101N or N-terminally acetylated G93A and D101N were examined by TEM of negatively stained samples (page 31, lines 5-7 up). G93A fibrils and D101N fibrils assembled from bacterial-purified G93A and D101N or N-terminally acetylated G93A and D101N were produced as described above (page 32, lines 7-8 up). A total of 7,799 movies for G93A fibrils assembled from N-terminally acetylated G93A and 5,442 movies for D101N fibrils assembled from N-terminally acetylated D101N were collected in super-resolution mode at a nominal magnification of ×105,000 (physical pixel size, 0.824 Å) and a dose of 15.717 e⁻ Å⁻² s⁻¹ (page 33, lines 6-9). For the G93A fibril assembled from N-terminally acetylated G93A, 13,314 fibrils were picked manually from 7,799 micrographs, for the D101N fibril assembled from N-terminally acetylated D101N were picked manually from 5,442 micrographs, and 1024-, 686-pixel, and 400-pixel boxes were used to extract particles by a 90% overlap scheme for 2D classification. After several iterations of 2D classification, particles with the same morphology were picked out (page 35, lines 4-10). Expi293F suspension cells (catalog number A14635) were obtained from the Thermo Fisher Scientific (Gibco) and cultured in OPM-293 CD05 Medium (OPM), supplemented with 10% (v/v) OPM-293 ProFeed (OPM) in 5% CO₂ at 37 °C (page 36, lines 1-9).*

We have also added the following subsection titled “*Fibril seeds from N-terminally acetylated G93A and D101N are more cytotoxic to cultured cells than are wild-type SOD1 fibril seeds generated under the same conditions*” into the Results section of the revision.

We then investigated the influence of N-terminal acetylation of SOD1 on the cytotoxicity of G93A fibrils and D101N fibrils. SH-SY5Y neuroblastoma cells, HEK-293T cells, and HT-22 neuron cells were cultured for 1 day, then 20 mM Tris-HCl buffer (pH 7.4) containing 5 mM TCEP (control) or fibril seeds from N-terminally acetylated wild-type SOD1, G93A, or D101N was diluted into tissue culture medium, respectively, and the cells were cultured for 1.5 days and further investigated by an MTT reduction assay and a CCK8 reduction assay (Fig. 6A–F). The final concentration of fibril seeds from N-terminally acetylated wild-type SOD, G93A, and D101N using ELISA assay (Appendix Fig. S2A) was also slightly smaller than 10 μ M. Because the length of the fibrils may affect the results as well, NS-EM images of the input fibrils for G93A (Appendix Fig. S2E), D101N (Appendix Fig. S2F), and wild-type SOD1 (Appendix Fig. S2G) fibrils, formed by N-terminally acetylated SOD1 proteins, are shown. Abundant short fibrils with similar lengths were also observed (Appendix Fig. S2E–G). Notably, fibril seeds from N-terminally acetylated G93A mutant also exhibited significantly higher cytotoxicity to SH-SY5Y cells (Fig. 6A,B), HEK-293T cells (Fig. 6C,D), and HT-22 neuron cells (Fig. 6E,F) than fibril seeds from N-terminally acetylated wild-type SOD1 ($P = 0.016, 0.0081, 0.0477, 0.0126, 0.0026, \text{ and } 0.0173$, respectively). Surprisingly, fibril seeds from N-terminally acetylated D101N mutant showed significantly greater cytotoxicity to SH-SY5Y cells (Fig. 6A,B), HEK-293T cells (Fig. 6C,D), and HT-22 neuron cells (Fig. 6E,F) than the wild-type SOD1 fibril seeds ($P = 0.0143, 0.0182, 0.0458, 0.0189, 0.00017, \text{ and } 0.0107$, respectively). Together, the data showed that G93A fibrils and D101N fibrils, formed by N-terminally acetylated SOD1 mutants, are notably more toxic than are wild-type SOD1 fibrils generated under the same conditions.

Together, these data demonstrate that N-terminally acetylated G93A and D101N form distinct amyloid fibril structures. Both mutants exhibit surprisingly similar toxicity in neuronal cells, suggesting a possible role of N-terminal acetylation of

SOD1 in the cytotoxicity of G93A fibrils and D101N fibrils, contributing to ALS pathology.

(The above paragraphs, pages 18-19).

Figure 6. Fibril seeds from N-terminally acetylated G93A and D101N are more cytotoxic to cultured cells than are wild-type SOD1 fibril seeds generated under the same conditions.

(A–F) Cytotoxicity of fibril seeds from Expi293F cell-purified G93A and D101N to SH-SY5Y neuroblastoma cells (A and B), HEK-293T cells (C and D), or HT-22 neuron cells (E and F) assessed by the MTT assay (A, C, and E) and the CCK8 assay (B, D, and F) compared with that of fibril seeds from Expi293F cell-purified wild-type SOD1. The cells were cultured for 1 day, then 20 mM Tris-HCl buffer (pH 7.4) containing 0 μ M SOD1 fibril seeds, wild-type SOD1 fibril seeds, G93A fibril seeds,

or D101N fibril seeds was diluted into tissue culture medium, respectively, and the cells were further cultured for 1.5 days. The final concentration of fibril seeds from the mammalian cell-purified wild-type SOD, G93A, and D101N was slightly smaller than 10 μ M. The cell viability (%) (open red circles shown in scatter plots) is expressed as the mean \pm S.D. (with error bars) of values obtained in $n = 4$ (A to F) biologically independent experiments. SOD1 fibrils, $P = 0.0016, 0.0027, 0.000017, 0.0104, 0.000091,$ and 0.0065 (A to F); G93A fibrils, $P = 0.00013, 0.0000016, 0.00000036, 0.00010, 0.00016, 0.00061$ (black) and $0.016, 0.0081, 0.0477, 0.0126, 0.0026, 0.0173$ (red) (A to F); and D101N fibrils, $P = 0.00011, 0.0000076, 0.00000015, 0.00014, 0.0000050, 0.000079$ (black) and $0.0143, 0.0182, 0.0458, 0.0189, 0.00017, 0.0107$ (red) (A to F). Statistical analyses were performed using two-sided Student's t tests. Values of $P < 0.05$ indicate statistically significant differences. The following notation was used throughout: $*P < 0.05,$ $**P < 0.01,$ $***P < 0.001,$ and $****P < 0.0001$ relative to the control. Cells treated with 20 mM Tris-HCl buffer (pH 7.4) containing 5 mM TCEP for 1.5 days were used as a control.

Comment #2 • I think these comments are clear.

REPLY: We greatly respect the referee for the critical assessment. We have acknowledged our shortcomings of the previous version of our work. We have expanded on the limitations of our study with bacterial purified recombinant SOD1 molecules and have tested our basic conclusions in other experimental systems that contain the usually N-terminally acetylated SOD1 (Figs. 5, 6, and EV2C,D).

Referee #3:

Summary:

This manuscript reported two cryo-EM structures of amyloid fibrils formed by G93A and D101N mutants of SOD1 protein. These mutations give rise to amyloid fibrils with distinct structures compared to native SOD1 fibrils. By toxicity assays on the SH-SY5Y neuroblastoma cells and HEK-293T cells, they found G93A fibrils are significantly more toxic than those formed by D101N, which do not show a marked increase in toxicity compared to wild-type SOD1 fibrils. The biggest weakness of this manuscript is the structures are of in vitro aggregated SOD1 protein, and the

structures are very different from “three key region model” proposed from the study on the transgenic mice, therefore the relevance to the human ALS diseases remains unknown. Toxicity assay would be more convincing if it was done on neuron cells and more indicators were studied except for viability. Nevertheless, the quality of Cryo-EM structures is good and novel. The mutants they studied are important and the manuscript should be of interest to the ALS field.

We sincerely thank the referee for recognizing the significance of our work. The referee’s suggestion is very valuable for us to improve our manuscript. Right now, we have discussed the limitations using bacterial-purified proteins as well as the divergence from amyloid fibril structures formed by ALS-causing SOD1 mutants and the “three key region model”, as followed the advice of referee #3. *The following are the limitations using bacterial-purified proteins as well as the divergence from amyloid fibril structures formed by ALS-causing SOD1 mutants and the “three key region model”. First, we considered the impact of cofactors that might be present in patients and mouse models of ALS but not in our cell-free fibrillization studies. Notably, such cofactors might promote the inclusion of more N-terminal residues in the fibril core, as suggested for these mutants by Furukawa et al. (2010) and Bergh et al. (2015) and as are present in the wild-type SOD1 fibril core (Wang et al., 2022). Second, we considered the impact of N-terminally acetylated SOD1 mutants that are present in patients and mouse models of ALS but not in our cell-free fibrillization studies. We suggest that N-terminal acetylation of SOD1 mutants might promote the inclusion of more N-terminal residues in the fibril core. Although the SOD1 fibril structures determined here may not be identical to those that accumulate in patients with ALS and the same SOD1 mutations that are N-terminally acetylated, this work provides important initial insights into the potential structural underpinnings of how these SOD1 mutants might aggregate and cause cytotoxicity in ALS. We plan to determine cryo-EM structures of in vitro-generated amyloid fibrils from N-terminally acetylated G93A and D101N mutants and collect structural data on the G93A fibril and the D101N fibril purified from the brains of patients with ALS or ALS transgenic mice in the near future (pages 23-24).*

We also sincerely thank the referee for his (her) expert suggestion that we should perform toxicity assays on neuron cells!! Indeed, toxicity assay would be more convincing if it was done on neuron cells and more indicators were studied

except for viability. According to the advice of referee #3, we have now provided the toxicity assay data of amyloid fibrils from bacterial-purified SOD1 proteins and Expi293F cell-purified G93A, D101N, and wild-type SOD1 including in neuron cells (Figs. 4E,F and 6A–F, pages 60 and 62). We have added the following subsection titled “*Fibril seeds from N-terminally acetylated G93A and D101N are more cytotoxic to cultured cells than are wild-type SOD1 fibril seeds generated under the same conditions*” into the Results section of the revision.

We then investigated the influence of N-terminal acetylation of SOD1 on the cytotoxicity of G93A fibrils and D101N fibrils. SH-SY5Y neuroblastoma cells, HEK-293T cells, and HT-22 neuron cells were cultured for 1 day, then 20 mM Tris-HCl buffer (pH 7.4) containing 5 mM TCEP (control) or fibril seeds from N-terminally acetylated wild-type SOD1, G93A, or D101N was diluted into tissue culture medium, respectively, and the cells were cultured for 1.5 days and further investigated by an MTT reduction assay and a CCK8 reduction assay (Fig. 6A–F). The final concentration of fibril seeds from N-terminally acetylated wild-type SOD, G93A, and D101N using ELISA assay (Appendix Fig. S2A) was also slightly smaller than 10 μ M. Because the length of the fibrils may affect the results as well, NS-EM images of the input fibrils for G93A (Appendix Fig. S2E), D101N (Appendix Fig. S2F), and wild-type SOD1 (Appendix Fig. S2G) fibrils, formed by N-terminally acetylated SOD1 proteins, are shown. Abundant short fibrils with similar lengths were also observed (Appendix Fig. S2E–G). Notably, fibril seeds from N-terminally acetylated G93A mutant also exhibited significantly higher cytotoxicity to SH-SY5Y cells (Fig. 6A,B), HEK-293T cells (Fig. 6C,D), and HT-22 neuron cells (Fig. 6E,F) than fibril seeds from N-terminally acetylated wild-type SOD1 ($P = 0.016, 0.0081, 0.0477, 0.0126, 0.0026, \text{ and } 0.0173$, respectively). Surprisingly, fibril seeds from N-terminally acetylated D101N mutant showed significantly greater cytotoxicity to SH-SY5Y cells (Fig. 6A,B), HEK-293T cells (Fig. 6C,D), and HT-22 neuron cells (Fig. 6E,F) than the wild-type SOD1 fibril seeds ($P = 0.0143, 0.0182, 0.0458, 0.0189, 0.00017, \text{ and } 0.0107$, respectively). Together, the data showed that G93A fibrils and D101N fibrils, formed by N-terminally acetylated SOD1 mutants, are notably more toxic than are wild-type SOD1 fibrils generated under the same conditions.

Together, these data demonstrate that N-terminally acetylated G93A and D101N form distinct amyloid fibril structures. Both mutants exhibit surprisingly similar toxicity in neuronal cells, suggesting a possible role of N-terminal acetylation of SOD1 in the cytotoxicity of G93A fibrils and D101N fibrils, contributing to ALS pathology.

(The above paragraphs, pages 18-19).

Figure 6. Fibril seeds from N-terminally acetylated G93A and D101N are more cytotoxic to cultured cells than are wild-type SOD1 fibril seeds generated under the same conditions.

(A–F) Cytotoxicity of fibril seeds from Expi293F cell-purified G93A and D101N to SH-SY5Y neuroblastoma cells (A and B), HEK-293T cells (C and D), or HT-22 neuron cells (E and F) assessed by the MTT assay (A, C, and E) and the CCK8 assay

(B, D, and F) compared with that of fibril seeds from Expi293F cell-purified wild-type SOD1. The cells were cultured for 1 day, then 20 mM Tris-HCl buffer (pH 7.4) containing 0 μ M SOD1 fibril seeds, wild-type SOD1 fibril seeds, G93A fibril seeds, or D101N fibril seeds was diluted into tissue culture medium, respectively, and the cells were further cultured for 1.5 days. The final concentration of fibril seeds from the mammalian cell-purified wild-type SOD, G93A, and D101N was slightly smaller than 10 μ M. The cell viability (%) (open red circles shown in scatter plots) is expressed as the mean \pm S.D. (with error bars) of values obtained in $n = 4$ (A to F) biologically independent experiments. SOD1 fibrils, $P = 0.0016, 0.0027, 0.000017, 0.0104, 0.000091, \text{ and } 0.0065$ (A to F); G93A fibrils, $P = 0.00013, 0.0000016, 0.00000036, 0.00010, 0.00016, 0.00061$ (black) and $0.016, 0.0081, 0.0477, 0.0126, 0.0026, 0.0173$ (red) (A to F); and D101N fibrils, $P = 0.00011, 0.0000076, 0.00000015, 0.00014, 0.0000050, 0.000079$ (black) and $0.0143, 0.0182, 0.0458, 0.0189, 0.00017, 0.0107$ (red) (A to F). Statistical analyses were performed using two-sided Student's t tests. Values of $P < 0.05$ indicate statistically significant differences. The following notation was used throughout: $*P < 0.05$, $**P < 0.01$, $***P < 0.001$, and $****P < 0.0001$ relative to the control. Cells treated with 20 mM Tris-HCl buffer (pH 7.4) containing 5 mM TCEP for 1.5 days were used as a control.

We have revised and added the following sentences into the revised manuscript. *SH-SY5Y neuroblastoma cells, HEK-293T cells, and HT-22 neuron cells were cultured for 1 day, then 20 mM Tris-HCl buffer (pH 7.4) containing 5 mM TCEP (control), wild-type SOD1 fibril seeds, G93A fibril seeds, or D101N fibril seeds was diluted into tissue culture medium, respectively, and the cells were cultured for 1.5 days and further investigated by an MTT reduction assay and a CCK8 reduction assay (Fig. 4A–F) (page 15, lines 6-11). Notably, fibril seeds from bacterial-purified G93A mutant exhibited significantly higher cytotoxicity to SH-SY5Y cells (Fig. 4A,B), HEK-293T cells (Fig. 4C,D), and HT-22 neuron cells (Fig. 4E,F) than fibril seeds from bacterial-purified wild-type SOD1 ($P = 0.010, 0.00013, 0.0219, 0.0446, 0.0057, \text{ and } 0.0063$, respectively). Fibril seeds from bacterial-purified D101N mutant, however, did not show significantly greater cytotoxicity to SH-SY5Y cells (Fig. 4A,B), HEK-293T cells (Fig. 4C,D), and HT-22 neuron cells (Fig. 4E,F) than the wild-type SOD1 fibril seeds ($P = 0.349, 0.0953, 0.523, 0.536, 0.612, \text{ and } 0.189$, respectively). Together, the data showed that G93A fibrils are notably more toxic*

while D101N fibrils are not significantly more toxic than wild-type SOD1 fibrils (pages 15-16).

Figure 4. Fibril seeds from bacterial-purified G93A are more cytotoxic to cultured cells than are wild-type SOD1 fibril seeds generated under the same conditions.

(A–F) Cytotoxicity of fibril seeds from bacterial-purified G93A and D101N to SH-SY5Y neuroblastoma cells (A and B), HEK-293T cells (C and D), or HT-22 neuron cells (E and F) assessed by the MTT assay (A, C, and E) and the CCK8 assay (B, D, and F) compared with that of fibril seeds from bacterial-purified wild-type SOD1. The cells were cultured for 1 day, then 20 mM Tris-HCl buffer (pH 7.4) containing 0 μ M SOD1 fibril seeds, wild-type SOD1 fibril seeds, G93A fibril seeds, or D101N fibril

seeds was diluted into tissue culture medium, respectively, and the cells were further cultured for 1.5 days. The final concentration of fibril seeds from bacterial-purified wild-type SOD, G93A, and D101N was slightly smaller than 10 μ M. The cell viability (%) (open red circles shown in scatter plots) is expressed as the mean \pm S.D. (with error bars) of values obtained in $n = 4$ (A to F) biologically independent experiments. SOD1 fibrils, $P = 0.0062, 0.00020, 0.0347, 0.0064, 0.00085, \text{ and } 0.0067$ (A to F); G93A fibrils, $P = 0.00010, 0.0000078, 0.00070, 0.00070, 0.000072, 0.00014$ (black) and $0.010, 0.00013, 0.0219, 0.0446, 0.0057, 0.0063$ (red) (A to F); and D101N fibrils, $P = 0.0086, 0.00041, 0.0093, 0.0043, 0.0045, 0.0037$ (black) and $0.349, 0.0953, 0.523, 0.536, 0.612, 0.189$ (red) (A to F). Statistical analyses were performed using two-sided Student's t tests. Values of $P < 0.05$ indicate statistically significant differences. The following notation was used throughout: $*P < 0.05$, $**P < 0.01$, $***P < 0.001$, and $****P < 0.0001$ relative to the control. n.s., not significant. Cells treated with 20 mM Tris-HCl buffer (pH 7.4) containing 5 mM TCEP for 1.5 days were used as a control.

Here are the details for the revision:

Comment #1 • The SDS-PAGE gels of SOD1 protein before and after in vitro aggregation should be added to show the intact of the protein, because it raised concern that the protein was degraded since only C-terminal protein was incorporated in the ordered core of structures and no proteinase inhibitors were used during apo-SOD1 protein purification.

REPLY: According to the advice of the referee, we have now added the SDS-PAGE analysis of SOD1 protein before and after in vitro aggregation into the revision, to show that full-length SOD1 was incorporated into the SOD1 fibrils (Appendix Fig. S1, page 4, Appendix Figures and Tables). We have revised and added the following sentences into the Results and Discussion sections of the revision, as followed referee's suggestion. *The SDS-PAGE gels of SOD1 protein before (G93A monomer and D101N monomer) and after in vitro aggregation (G93A fibril and D101N fibril) have been added to show the intact of the protein (Appendix Fig. S1). The SDS-PAGE experiments showed that the protein was not degraded (Appendix Fig. S1) though no proteinase inhibitors were used during apo-SOD1 protein purification, but still only C-terminal protein was*

incorporated in the ordered core of structures (Figs. 2 and 3) (pages 11-12). Because it raised concern that the protein was degraded since only C-terminal protein was incorporated in the ordered core of structures, the SDS-PAGE gels of SOD1 protein before and after in vitro aggregation have been added to show that full-length SOD1 was incorporated into the SOD1 fibrils (page 23, lines 9-12 up).

Appendix Figure S1. The SDS-PAGE gels of SOD1 protein before (G93A monomer and D101N monomer) and after *in vitro* aggregation (G93A fibril and D101N fibril) have been added to show the intact of the protein.

(A–C) All SDS-PAGE experiments were repeated three times and the results were reproducible. The samples of G93A dimers, D101N dimers, G93A fibrils, and D101N fibrils were dissolved in 8 M urea and separated by 12.5% SDS-PAGE. The gels were stained with Coomassie Blue staining solution and washed with destaining buffer. The SDS-PAGE experiments show that the protein was not degraded, although only C-terminal protein was incorporated in the ordered core of structures and no proteinase inhibitors were used during apo-SOD1 protein purification.

We have added the following subsection titled “Coomassie Blue staining” (page 31) into the Methods section of the revision. *The samples of G93A dimers, D101N dimers, G93A fibrils, and D101N fibrils were dissolved in 8 M urea for more than 2 hours. The samples were then boiled in SDS-PAGE loading buffer for 15 min and separated by 12.5% SDS-PAGE. The gels were stained with Coomassie Blue staining solution for 2 hours and then washed with destaining buffer until the band of SOD1 protein appeared (page 31, lines 9-13).*

Comment #2 • The accuracy of using NanoDrop to measure the concentration of the fibrils is questionable. ELISA or MSD assays could be used for accurately measuring the concentration of monomers denatured from the fibrils. The length of the fibrils may affect the results as well. NS-EM images of the input fibrils for WT, G93A and D101N fibrils should be shown.

REPLY: We sincerely thank the referee for his (her) expert suggestions. We totally agree that the accuracy of using NanoDrop to measure the concentration of the fibrils is questionable and that the length of the fibrils may affect the results as well. According to the advice of referee #3, we have now used ELISA assay to accurately measure the concentration of monomers denatured from the SOD1 fibrils (Appendix Fig. S2A, page 5, Appendix Figures and Tables) and have shown negative-stain electron microscopy (NS-EM) images of the input fibrils for G93A, D101N, and wild-type SOD1 fibrils (Appendix Fig. S2B–G, page 5, Appendix Figures and Tables). We have revised and added the following sentences into the Results and Discussion sections of the revision, as followed referee's suggestions. *ELISA assay was used for accurately measuring the concentration of monomers denatured from the SOD1 fibrils (Appendix Fig. S2A), and the final concentration of fibril seeds from bacterial-purified wild-type SOD, G93A, and D101N using ELISA assay was slightly smaller than 10 μ M. Because the length of the fibrils may affect the results as well, negative-stain electron microscopy (NS-EM) images of the input fibrils for G93A (Appendix Fig. S2B), D101N (Appendix Fig. S2C), and wild-type SOD1 (Appendix Fig. S2D) fibrils are shown. Abundant short fibrils with similar lengths were observed (Appendix Fig. S2B–D) (page 15, lines 5-12 up). The final concentration of fibril seeds from N-terminally acetylated wild-type SOD, G93A, and D101N using ELISA assay (Appendix Fig. S2A) was also slightly smaller than 10 μ M. Because the length of the fibrils may affect the results as well, NS-EM images of the input fibrils for G93A (Appendix Fig. S2E), D101N (Appendix Fig. S2F), and wild-type SOD1 (Appendix Fig. S2G) fibrils, formed by N-terminally acetylated SOD1 proteins, are shown. Abundant short fibrils with similar lengths were also observed (Appendix Fig. S2E–G) (pages 18-19).*

Appendix Figure S2. Accurate measurement of the concentration of monomers denatured from the SOD1 fibrils (A) and NS-EM images of the input fibrils for G93A, D101N and wild-type SOD1 fibrils (B–G).

(A) The samples of G93A fibrils, D101N fibrils, and wild-type SOD1 fibrils were dissolved in 8 M urea and then ELISA assay was used for accurately measuring the concentration of monomers denatured from the SOD1 fibrils. The concentration of fibril seeds from bacterial-purified SOD1 proteins and the mammalian cell-purified SOD1 proteins, measured by ELISA assay, was 0.667–0.714 mg/ml and 0.692–0.702 mg/ml, respectively, slightly smaller than that measured using NanoDrop (0.72 mg/ml). (B–G) Negative-stain electron microscopy (NS-EM) images of the input fibrils for G93A (B and E), D101N (C and F), and wild-type SOD1 (D and G) fibrils formed by bacterial-purified SOD1 proteins (B–D) and by the mammalian cell-purified SOD1 proteins (E–G). Abundant short fibrils with similar lengths were observed. Scale bars, 200 nm.

Comment #3 • The orientation of the G93A structure in Fig S4 is different from that in Fig 2, which may cause confusion. It should be flipped 180° along X-axis.

REPLY: We apologize for this confusion. Indeed, the orientation of the G93A structure in the previous Fig. S4 is different from that in Fig. 2, which may cause confusion. The orientation of the structure of G93A fibril in (A and B) has been flipped 180° along X-axis so that it is the same as that in Fig. 2 (page 71, lines 3-4, Legend of Fig. EV5).

Figure EV5. Close-up view of the stick representation of the structure of G93A fibril stabilized by an intramolecular salt bridge and two hydrogen bonds.

(A) A space-filled model overlaid onto a stick representation of the G93A fibril in which a single protofilament containing five molecular layers is shown in light green. The ALS-causing mutation site Ala93 is highlighted in yellow. Asp96/Arg143 pairs that form a new salt bridge are highlighted in red (oxygen atom in Asp96) and blue (nitrogen atom in Arg143), and the salt bridge region is magnified in (B). Asn139/Arg143 pairs and Gln153/Arg143 pairs that form hydrogen bonds are highlighted in red (oxygen atoms in Asn139 and Gln153) and blue (two nitrogen atoms in Arg143), and two hydrogen bond regions are also magnified in (B). (B) A magnified top view of the salt bridge region of a G93A protofilament, where Asp96/Arg143 pairs form a salt bridge. A side view (right) highlighting a strong salt bridge between Arg143 and Asp96 from the same molecular layer, with a distance of 2.7 Å (blue). Magnified top views of the two hydrogen bond regions of a G93A protofilament, where two pairs of amino acids (Asn139 and Arg143, and Gln153 and Arg143) form two hydrogen bonds. Two side views (right) highlighting a hydrogen bond between Arg143 and Asn139 from the same molecular layer, with a distance of 3.6 Å (blue), or between Arg143 from the molecular layer (i) and Gln153 from the adjacent molecular layer ($i - 1$), with a distance of 2.7 Å (red). The orientation of the

structure of G93A fibril in (A and B) has been flipped 180° along X-axis so that it is the same as that in Fig. 2.

Comment #4 • For each structure, separate model refinements should be performed against a single half-map, and the resulting model should be compared with the other half-map to confirm the absence of overfitting.

REPLY: According to the advice of referee #3, we have now performed separate model refinements against a single half-map for each structure and have compared the resulting model with the other half-map to confirm the absence of overfitting (Fig. EV4, page 69). We have revised and added the following sentences into the Results section of the revision, as followed referee's suggestion. *For each structure of the G93A fibril and the D101N fibril, separate model refinements were performed against a single half-map, and the resulting model was compared with the other half-map to confirm the absence of overfitting (Fig. EV4A–D). The comparison of the two optimized half-maps, half1 map and half2 map, from the G93A fibril (Fig. EV4A) and the D101N fibril (Fig. EV4B) 3D auto-refine process without mask demonstrates that the two half-maps match well (page 12, lines 8-13).*

Figure EV4. For each structure of G93A fibril and D101N fibril, separate model refinements were performed against a single half-map, and the resulting model was compared with the other half-map to confirm the absence of overfitting.

(A and B) Comparison of the two optimized half-maps, half1 map (green and purple, respectively) and half2 map (cyan and magenta, respectively), from the G93A fibril (A) and the D101N fibril (B) 3D auto-refine process without mask, and the two half-maps match well. (C) Comparison of half1 map (green) and half2 map (cyan) of the G93A fibril with the atomic model overlaid. (D) Comparison of half1 map (purple) and half2 map (magenta) of the D101N fibril with the atomic model overlaid.

Comment #5 • The cross-sectional view of 3D map in Fig 1A and B could be improved by showing the cross-sectional view of one rung.

The indicated pitch range indicated in Figure 1E is not accurate.

The colour bar in Fig 3G is reversed.

Positions of G93A and D101N should be labelled in Fig 5B,C,D.

REPLY: Thanks the referee for the comment concerning Fig. 1A,B. The referee is correct. According to the advice of referee #3, we have now improved the cross-sectional view of 3D map in Fig. 1A,B by showing the cross-sectional view of one rung (Fig. 1A,B, page 57). (A and B) Cross-sectional view of the 3D map of the G93A fibril (A) or the D101N fibril (B) showing a protofilament comprising a C-terminal segment (green) and improving by showing the cross-sectional view of one rung (page 49, lines 5-7, Legend of Fig. 1).

Figure 1. Comparison of the cryo-EM structures of the G93A fibril and the D101N fibril.

(A and B) Cross-sectional view of the 3D map of the G93A fibril (A) or the D101N fibril (B) showing a protofilament comprising a C-terminal segment (green) and improving by showing the cross-sectional view of one rung. (C) Cross-sectional view of the 3D map of the wild-type SOD1 fibril showing a protofilament comprising not only a C-terminal segment (green) but also an N-terminal segment (yellow) with an unstructured flexible fragment (magenta dashed line) (Wang et al, 2022). Scale bars, 5 nm. For full clarity, we false color the equivalent regions in panels A, B, and C.

We apologize for this mistake. We have now adjusted the pitch range in Fig. 1E (page 57) to make sure that the indicated pitch range is accurate, as followed referee #3's nice suggestion. We have adjusted the pitch range shown in E to make sure that the indicated pitch range is accurate (page 49, lines 7-8 up, Legend of Fig. 1).

Figure 1. Comparison of the cryo-EM structures of the G93A fibril and the D101N fibril.

(**D** and **E**) 3D map of the G93A fibril (**D**) or the D101N fibril (**E**) showing a single protofilament (in light green for **D** and light purple for **E**) intertwined into a left-handed helix, with a fibril core width of ~8.1 (**D**) or ~6.3 (**E**) nm and a half-helical pitch of 120.3 (**D**) or a helical pitch of 64.2 (**E**) nm (left). We have adjusted the pitch range shown in **E** to make sure that the indicated pitch range is accurate. Enlarged section of the G93A fibril (**D**) or the D101N fibril (**E**) showing a side view of the density map (top right). Close-up view of the density map on the left showing that the subunit in a protofilament stacks along the fibril axis with a helical rise of 4.88 (**D**) or 4.81 (**E**) Å (top right). Top view of the density map of the G93A fibril (**D**) or the D101N fibril (**E**) (bottom right).

We apologize for this mistake. We have now reversed the color bar in Fig. 3G (page 59) to make sure that the indicated color bar is accurate, as followed referee #3's suggestion. We have reversed the color bar shown in G to make sure that the indicated color bar is accurate (page 51, lines 1-2 up, Legend of Fig. 3).

Figure 3. The ALS-causing SOD1 mutant D101N also forms a novel amyloid fibril structure.

(G) Hydrophobic surface representation of the structure of a D101N fibril core shown in (D). The surface of the D101N fibril core is shown according to the electrostatic properties (red, negatively charged; blue, positively charged) (F) or the hydrophobicity (yellow, hydrophobic; blue, hydrophilic) (G) of the residues. We have reversed the color bar shown in G to make sure that the indicated color bar is accurate.

Thanks the referee for the comment concerning the previous Fig. 5B–D. The referee is correct. According to the advice of referee #3, we have now labelled the positions of G93A and D101N in Fig. 7B–D (page 63). *Positions of G93A (magenta) and D101N (blue) are labelled in (B–D) (page 56, lines 1-2, Legend of Fig. 7).*

Figure 7. Comparison of the structures of the apo form of SOD1, the wild-type SOD1 fibril, the G93A fibril, and the D101N fibril.

(**B–D**) Ribbon representation of the structures of a wild-type SOD1 fibril core (**B**), a G93A fibril core (**C**), and a D101N fibril core (**D**), all of which contain one molecular layer and a monomer. Positions of G93A (magenta) and D101N (blue) are labelled in (**B–D**). (**E**) Overlay of the structures of a wild-type SOD1 fibril core (orange), a G93A fibril core (magenta), and a D101N fibril core (blue).

Dear Prof. Liang

Thank you for the submission of your revised manuscript to EMBO reports. We have now received the full set of referee reports that is copied below.

As you will see, the referees acknowledge that the revision has strengthened the manuscript. While referee #3 points out that the structures are limited to bacterially expressed SOD1 proteins, both referees consider the structural and toxicity information of value for the field. We will therefore proceed with publication, pending that the remaining concerns from referee #3 are adequately addressed, in particular point 4, the Overfitting Assessment.

From the editorial side, there are also a few things that we need before we can proceed with the official acceptance of your study.

- Please remove the figures from the manuscript text file. Only the legends of both main and EV figures should be provided at the end of the manuscript.
- Regarding the Author Contributions, we now use CRediT to specify the contributions of each author in the journal submission system. Therefore, please remove the Author Contributions from the manuscript file and make sure that the author contributions in our online manuscript tracking system are correct and up-to-date. The information you specified in the system will be automatically retrieved and typeset into the article. You can enter additional information in the free text box provided, if you wish.
- Please remove the funders and grants in the Comments box of our online manuscript tracking system and enter them as separate funders via the More Funders option.
- Please provide callouts for Figure 2C and Figure 3CF in the manuscript text.
- Please provide the Appendix file as PDF. The title page should only have the title "Appendix" and a table of content with page numbers. In addition, the Appendix needs to be clean, i.e., without colored font, highlighting, etc. It will not be typeset.
- Please remove the Reagents and Tools table from the manuscript and upload it as separate file (file type Reagents and Tools table).
- Data availability section:
 - 1) Please remove the first sentence. The section should only refer to data deposited on public repositories.
 - 2) Please provide specific URLs for EMD 60996, EMD 60998, PDB 9IYD, and PDB 9IYJ datasets. I.e., the URL should resolve to the dataset, not just the database.
 - 3) Please deposit the mass spectrometry data in a public repository and provide the accession code and link in the Data Availability section as well.
- We perform a routine image and data integrity analysis on all revised manuscript and it came to our attention that the image shown in Figure 1C has been reused from a published article (WT SOD1 fibril). The same image appears to be shown in Figure 1B of Nat Commun. 2022 Jun 17;13:3491. doi: 10.1038/s41467-022-31240-4
This reuse has to be clearly called out in the relevant section of the results and in the figure legend, i.e., that you compare the 3D map of the here analysed SOD1 G83A and D101N to previously published data on the WT SOD1. The figure legend can state: Figure 1C is reused between..."
- Appendix Figure S1 - A,B,C have low resolution. Please provide an updated high resolution figure.
- We perform a routine plagiarism check on all revised manuscripts and noticed that some paragraphs are either identical or very similar to text in your previous paper (Wang et al, Science Advances 2024).
 - 1) Page 6, starting from "Recently, we reported a cryo-electron microscopy" to "... (TDP-43) in neurons is another pathological feature of ALS"
 - 2) Page 9 starting from "Third, injection of in vitro-generated amyloid fibrils..." to "...and cause the familial form of ALS."
 - 3) Page 9 to 11, the first three paragraphs of the results.
 - 4) Page 21 starting from "... and provides high-resolution cryo-EM structures..." to "... We address the relevance of the in vitro-prepared amyloid fibrils to human diseases below."
 - 5) Page 24Please rephrase these section to avoid any ambiguities.
- Finally, EMBO Reports papers are accompanied online by
 - A) a short (1-2 sentences) summary of the findings and their significance,
 - B) 2-3 bullet points highlighting key results and

C) a schematic summary figure that provides a sketch of the major findings (not a data image). Please provide the summary figure as a separate file in PNG or JPG format at a size of 550x300-600 pixels (width x height). Please note that the size is rather small and that text needs to be readable at the final size. Please send us this information along with the revised manuscript.

With kind regards,

=====

Referee #2:

The authors have addressed my concerns in the revised manuscript. I am now happy to recommend publication of this work.

Referee #3:

Zhang et al. have submitted a revision that directly tackles the main concern of their cytotoxicity assays-by adding experiments in Expi293F cells to compare G93A and D101N fibrils against wild-type fibrils under identical conditions, and by extending the assays to HT-22 neurons. It's impressive they managed this in such a short timeframe. However, it also brings up new questions about how these findings really advance the field.

1. Core Finding vs. Revised Context

The heart of the paper is the Cryo-EM structures of in vitro fibrils formed by bacterial-purified SOD1 mutants G93A and D101N. Yet their revision hints that fibrils formed by mammalian-expressed, N-terminally acetylated proteins adopt completely different conformations from those made by proteins purified from bacteria. I appreciate that they plan to solve and publish the acetylated-protein structures separately, but unfortunately it downplays the importance of the bacterial-derived SOD1 mutant structures here. They also barely discuss how these structural differences relate to cytotoxicity.

2. Missing Wild-Type 2D Classifications

They now show EM images of fibrils from Expi293F-expressed SOD1 (wild-type, G93A, D101N), yet surprisingly they omit 2D class averages for the wild-type fibrils while including them for both mutants. For a proper side-by-side comparison, those wild-type classifications would be good to be included.

3. Fibril Length and Sonication Details

Their ELISA assays accurately quantify monomers released from fibrils, and Appendix Figure S2 shows that all fibrils are similar in length-nice work. But these fibrils look shorter than those in Figure EV1, suggesting they may have been sonicated before the cytotoxicity tests. That step should be described in the Methods. Also, the legend for Appendix Figure S1 doesn't clearly explain what distinguishes panels A, B, and C.

4. Overfitting Assessment

They answered the overfitting concern, but the response isn't up to standard. In Cryo-EM work, the conventional check for overfitting follows the approach in Fig. S2D of Science (2014), 343, 1485-1489. They should apply that method to convincingly demonstrate the absence of overfitting.

Referee #2:

Summary:

The authors have addressed my concerns in the revised manuscript. I am now happy to recommend publication of this work.

We sincerely thank the referee for recognizing the significance of our work. The referee's suggestion is very valuable for us to improve our manuscript.

Referee #3:

Summary:

Zhang et al. have submitted a revision that directly tackles the main concern of their cytotoxicity assays-by adding experiments in Expi293F cells to compare G93A and D101N fibrils against wild-type fibrils under identical conditions, and by extending the assays to HT-22 neurons. It's impressive they managed this in such a short timeframe.

REPLY: We sincerely appreciate the time and thoughtful feedback you have dedicated to reviewing our manuscript. Your positive evaluation of the significance of our research is deeply encouraging. We have carefully considered all the reviewers' valuable suggestions and have incorporated them into the revised manuscript.

However, it also brings up new questions about how these findings really advance the field.

Comment #1 • Core Finding vs. Revised Context

The heart of the paper is the Cryo-EM structures of in vitro fibrils formed by bacterial-purified SOD1 mutants G93A and D101N. Yet their revision hints that fibrils formed by mammalian-expressed, N-terminally acetylated proteins adopt completely different conformations from those made by proteins purified from bacteria. I appreciate that they plan to solve and publish the acetylated-protein structures separately, but unfortunately it downplays the importance of the bacterial-derived SOD1 mutant structures here. They also barely discuss how these structural differences relate to cytotoxicity.

REPLY: We sincerely appreciate the referee's feedback and apologize for the unintended shift in the manuscript's focus. As another reviewer pointed out that the proteins we used were expressed in Escherichia coli where the N-terminus is

not acetylated, which was the fatal limitation of our initial work, we rigorously addressed this concern by repeating the work in the suggested background - a eukaryote that acetylates the N terminus after removal of Met 1. We have to admit that this revision unavoidably dilutes some of the original research priorities and we regret any disruption to the coherence of narrative. However, we maintain that the core contribution of this paper is still based on the bacterial-derived SOD1 mutant structures, while findings regarding N-terminally acetylated proteins are merely preliminary and should serve as an introduction to new directions which will require extensive future work.

Comment #2 • Missing Wild-Type 2D Classifications

They now show EM images of fibrils from Expi293F-expressed SOD1 (wild-type, G93A, D101N), yet surprisingly they omit 2D class averages for the wild-type fibrils while including them for both mutants. For a proper side-by-side comparison, those wild-type classifications would be good to be included.

REPLY: We sincerely thank the referee for this important suggestion that we should provide the missing wild-type 2D classifications. In response to the referee's request, we have now provided the most representative images of the 2D class averages for the wild-type fibrils in Appendix Fig. S3, with the understanding that further optimization and refinement will be required to produce better results (Appendix Fig. S3, page 5, Appendix Figures and Tables), yet the current wild-type 2D classifications are not good enough for 3D classifications. We have now revised the two sentences of our article as "*The cryo-EM micrographs (Fig. 5G–I) and 2D class average images obtained using RELION3.1 (Scheres, 2020) showed that the G93A fibril and the D101N fibril formed by N-terminally acetylated SOD1 mutants and the wild-type fibril formed by N-terminally acetylated wild-type SOD1 were composed of a single protofilament less intertwined (Fig. EV2C) and intertwined (Fig. EV2D and Appendix Fig. S3), respectively. The 2D class average images showed that at least in vitro, N-terminally acetylated G93A and D101N form fibril conformers that differ from each other and from those formed from bacterial-purified G93A and D101N (Fig. EV2A–D) as well as N-terminally acetylated wild-type SOD1 (Appendix Fig. S3)*" (page 18, lines 4-13), as followed referee's suggestions.

Expi293F cell-purified wild-type SOD1

Appendix Figure S3. Cryo-EM images of wild-type SOD1 fibril.

Reference-free 2D class averages of the wild-type SOD1 fibril formed by Expi293F cell-purified wild-type SOD1 showing a single protofilament intertwined. Scale bar, 10 nm.

Comment #3 • Fibril Length and Sonication Details

Their ELISA assays accurately quantify monomers released from fibrils, and Appendix Figure S2 shows that all fibrils are similar in length-nice work. But these fibrils look shorter than those in Figure EV1, suggesting they may have been sonicated before the cytotoxicity tests. That step should be described in the Methods. Also, the legend for Appendix Figure S1 doesn't clearly explain what distinguishes panels A, B, and C.

REPLY: We apologize for such an ambiguity in our wording due to oversight. Just as your correctly suggested, we sonicated the fibrils before the cytotoxicity tests, to achieve a homogeneous reduction in length and thereby enhancing cellular uptake efficiency. We have updated the Methods section accordingly based on your valuable feedback as follows. *We sonicated the SOD1 fibrils for 5 min (5 s on, 5 s off) on ice before the cytotoxicity tests so that all fibrils are similar in length (Methods, page 30, lines 6-7 up). Our ELISA assays accurately quantify monomers released from fibrils, and Appendix Fig. S2 shows that all fibrils are similar in length (page 16, lines 1-3).*

We sincerely thank referee #3 for this important suggestion that we should clearly explain what distinguishes panels A, B, and C in the legend for Appendix Fig. S1!! Right now, we have carefully rephrased the legend for Appendix Fig. S1 to clearly explain what distinguishes panels A, B, and C, as followed referee's

suggestions. (A) SDS-PAGE analysis of SOD1 protein before and after *in vitro* aggregation have been added. B and C represent two of the biological replicates of A (Legend of Appendix Fig. S1, page 3, Appendix Figures and Tables).

Appendix Figure S1. The SDS-PAGE gels of SOD1 protein before (G93A monomer and D101N monomer) and after *in vitro* aggregation (G93A fibril and D101N fibril) have been added to show the intact of the protein.

(A) SDS-PAGE analysis of SOD1 protein before and after *in vitro* aggregation have been added. In brief, the samples of G93A dimers, D101N dimers, G93A fibrils, and D101N fibrils were dissolved in 8 M urea and separated by 12.5% SDS-PAGE. The gels were stained with Coomassie Blue staining solution and washed with destaining buffer. All SDS-PAGE experiments were repeated three times and the results were reproducible. B and C represent two of the biological replicates of A. Marks at the left of the gels indicate the positions of the molecular weight markers. The SDS-PAGE experiments show that the protein was not degraded though no proteinase inhibitors were used during apo-SOD1 protein purification.

Comment #4 • Overfitting Assessment

They answered the overfitting concern, but the response isn't up to standard. In Cryo-EM work, the conventional check for overfitting follows the approach in Fig. S2D of Science (2014), 343, 1485-1489. They should apply that method to convincingly demonstrate the absence of overfitting.

REPLY: We sincerely apologize for the misunderstanding in our initial submission and greatly appreciate your constructive feedback. As you rightly pointed out, we have now rigorously re-evaluated our model for potential overfitting by comprehensive validation program in PHENIX 1.15.2. The revised results (Figure EV4, page 59) confirm the robustness of our model without overfitting. Thank you again for your valuable critique, which has strengthened our work. We have added the following sentences into the Results section of the revision, as followed referee's suggestion. *Using the approach by Amunts et al. (2014), we compared the FSC curves between the final refined model and the map reconstructed from all fibrils, the FSC curves between a model refined against the first half of the two independent half maps used for gold-standard FSC versus the*

reconstruction from that same half, and the FSC curves between a model refined against the first half of the two independent half maps versus the second independent half map, and a good superimposition of the two independent halves was observed (Fig. EV4A,B). The vertical lines at 3.09 Å and 2.92 Å indicate the highest resolution used in model refinement of the G93A fibril and the D101N fibril, respectively (Fig. EV4A,B). These data convincingly demonstrate the absence of overfitting (page 12, lines 5-14). Accordingly, one related publication (Amunts A, Brown A, Bai XC, Llácer JL, Hussain T, Emsley P, Long F, Murshudov G, Scheres SHW, Ramakrishnan V (2014) Structure of the yeast mitochondrial large ribosomal subunit. *Science* 343:1485–1489. pages 39-40) has been added into the revision.

Figure EV4. For each structure of G93A fibril and D101N fibril, separate model refinements were performed against a single half-map, and the resulting model was compared with the other half-map to confirm the absence of overfitting.

(A and B) Fourier shell correlation (FSC) curves between the density map and the model. The FSC curves between the final refined model and the map reconstructed from all fibrils (black curve); between a model refined against the first half of the two independent half maps used for gold-standard FSC *versus* the reconstruction from that same half (red curve); and between a model refined against the first half of the two independent half maps *versus* the second independent half map (blue curve). The vertical lines at 3.09 Å (A) and 2.92 Å (B) indicate the highest resolution used in model refinement of the G93A fibril and the D101N fibril, respectively. These data convincingly demonstrate the absence of overfitting.

Prof. Yi Liang
Wuhan University
College of Life Sciences
Hubei Key Laboratory of Cell Homeostasis, College of Life Sciences, TaiKang Center for Life and Medical Sciences, Wuhan University, Wuhan 430072, China
Wuhan, Hubei 430072
China

Dear Prof. Liang,

I am very pleased to accept your manuscript for publication in the next available issue of EMBO reports. Thank you for your contribution to our journal.

Yours sincerely,
